# ONLINE TIME SERIES PREDICTION USING FEATURE ADJUSTMENT

**Xiannan Huang**
College of Transportation
Tongji University
Shanghai, 201804, China
`huang_xn@tongji.edu.cn`

**Shuhan Qiu**
College of Transportation
Tongji University
Shanghai, 201804, China
`qiusuan@tongji.edu.cn`

**Jiayuan Du**
College of Computer Science
Tongji University
Shanghai, 201804, China
`dujiayuan@tongji.edu.cn`

**Chao Yang** [*]
College of Transportation
Tongji University
Shanghai, 201804, China
`tongjiyc@tongji.edu.cn`

## ABSTRACT

Time series forecasting is of significant importance across various domains. However, it faces significant challenges due to distribution shift. This issue becomes particularly pronounced in online deployment scenarios where data arrives sequentially, requiring models to adapt continually to evolving patterns. Current time series online learning methods focus on two main aspects: selecting suitable parameters to update (e.g., final layer weights or adapter modules) and devising suitable update strategies (e.g., using recent batches, replay buffers, or averaged gradients). We challenge the conventional parameter selection approach, proposing that distribution shifts stem from changes in underlying latent factors influencing the data. Consequently, updating the feature representations of these latent factors may be more effective. To address the critical problem of delayed feedback in multi-step forecasting (where true values arrive much later than predictions), we introduce ADAPT-Z (Automatic Delta Adjustment via Persistent Tracking in Z-space). ADAPT-Z utilizes an adapter module that leverages current feature representations combined with historical gradient information to enable robust parameter updates despite the delay. Extensive experiments demonstrate that our method consistently outperforms standard base models without adaptation and surpasses state-of-the-art online learning approaches across multiple datasets.

## 1 INTRODUCTION

Accurate time series forecasting is critically important for numerous applications including traffic management (Kadiyala & Kumar, 2014), disease control (Morid et al., 2023), and many other domains. Significant research attention has been directed toward this field in recent years (Li & Law, 2024). However, a major challenge stems from the frequent occurrence of distribution shift in time series data, where patterns in test data are changing over time (Pham et al., 2023). Furthermore, since data arrives sequentially during deployment, adjusting a pre-trained model for online predictions becomes a particularly important problem.

The online time series prediction process in existing research can be divided into the following two steps: first, updating the model using newly acquired data, and then deploying this updated model at the next timestep. Upon arrival of new data at that subsequent timestep, the model is updated again and then applied to the following timestep, creating a repeating loop (Wen et al., 2023; Guo et al., 2024). A fundamental method within this framework is Online Gradient Descent (OGD). In OGD, upon observing a single data sample, the loss is computed and the model's parameters

---

[*]Corresponding Author.

are immediately updated based on this loss; the updated model is then used at the next timestep. Most existing online time series learning methods can be regraded as enhancements or variations of this fundamental OGD procedure. Consequently, the core challenges addressed by these methods involve two parts: determining which parameters to update and how to update them.

Regarding which parameters to update, some studies propose updating only the parameters of the final layer (Chen et al., 2024b). Additionally, many studies advocate using adapter structures, where the base model's parameters remain frozen, and only the smaller adapter modules– typically a few simple MLPs with fewer parameters – are updated (Lau et al., 2025; Guo et al., 2024). This approach yields more stable updates due to the reduced number of adjusted parameters. Concerning how the updates are performed, some methods use a small batch of recent samples (Lau et al., 2025) or employ strategies like a replay buffer, selecting similar samples from past stored data ("memory") (Chen et al., 2024b).

Therefore, the core challenge for online time series forecasting lies in identifying appropriate parameters and devising effective parameter update strategies. This raises a question: are the parameters typically chosen for updates in existing literature truly optimal? We propose a key insight: what appears as distribution shift on the surface likely stems from underlying latent factors. For instance, in a traffic flow prediction scenario, observed data might show vehicle counts at an intersection. However, the true influencing factors are complex latent variables—such as public preference for private vehicles, temperature, or economic conditions. Distribution shift often occurs because these latent factors change over time. Suppose the economy was prosperous during training but declined during testing – this could reduce people's willingness to drive private cars, leading to lower traffic volumes. Thus, the crucial parameters for update might be features capturing these hidden factors. Besides, some studies suggest that for complex deep learning models, the model can be conceptually divided into an encoder and a prediction head. The output from the encoder represents the underlying factors (or their combinations) causing the observed time series (Li et al., 2025). As a result, modifying features at this level is more intuitively aligned with the root causes of distribution drift.

The next question is: how to update these features? The simplest approach is gradient-based methods. However, a significant challenge arises when making multi-step forecasts. For example, when predicting the next 24 steps, the true value for a forecast made at time $t$ is not available until time $t + 24$. Therefore, the error (and consequently, the gradient) calculated at time $t$ must use the forecast made 24 steps earlier ($t - 24$) (Lau et al., 2025). This delayed feedback can lead to unreliable gradients. To address this limitation, we propose leveraging an adapter module that utilizes both the current feature representations and past gradients to update parameters effectively. We named our proposed method as ADAPT-Z (Automatic Delta Adjustment via Persistent Tracking in Z-space) and summary the main difference between our method and existing baselines in Figure 1.

We summarize our contributions as follows:

**(1) Feature-Space Adaptation Paradigm.** We identify that distribution shifts in online prediction stem from changes in latent variables governing data distribution. Unlike existing methods that update model parameters, we propose to address this at the feature level, introducing a new adaptation paradigm for online time-series forecasting.

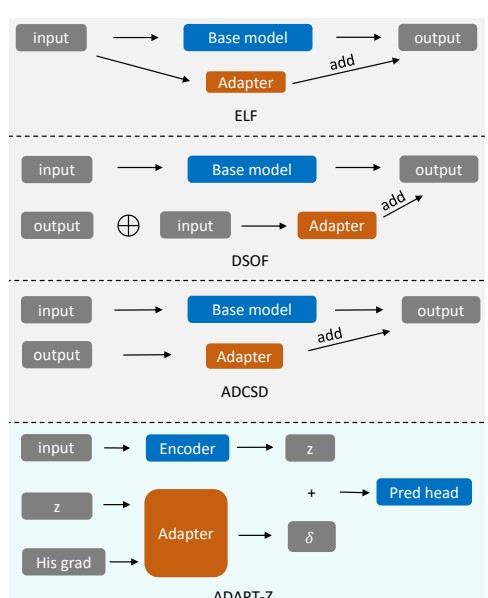

Figure 1: The difference between Our method (ADAPT-Z) and other methods

**(2) Simplicity Meets Effectiveness.** We demonstrate that even the simplest online gradient descent algorithm at the feature level performs comparably to, or better than, complex parameter-update methods (Section 4.3.1), challenging the convention that sophisticated adaptation mechanisms are necessary.

**(3) ADAPT-Z Method.** We propose ADAPT-Z, a lightweight online adaptation method that uses an MLP to update features by fusing current representations with historical gradients. This design addresses the delayed feedback problem in multi-step forecasting while achieving state-of-the-art results across multiple base models and datasets.

**(4) Learn-to-Adapt Phenomenon.** We discover that pre-training with feature gradients enables models to adapt to distribution shifts during deployment without any parameter updates, exhibiting a "learn-to-adapt" effect (Section 4.3.2).

## 2 RELATED WORK

As mentioned before, the research about online time series forecasting can be summarized according to two aspects: determining which parameters to update and how to update these parameters.

**Choosing parameters to update:** Different online learning method updates different parameters. For example, OneNet (Wen et al., 2023) updates the ensemble weights for two models, FSNet (Pham et al., 2023) updates convolutional layer weights and feature modification weights, while the DSOF (Lau et al., 2025) updates only the student model's weights. Notably, most literature avoids updating all model parameters. Instead, a common strategy involves introducing a small adapter module (Lau et al., 2025; Guo et al., 2024; Lee et al., 2025; Medeiros et al., 2025; Grover & Etemad, 2025; Kim et al., 2025) for online adaptation. Updating only a limited parameter subset within such an adapter significantly enhances learning stability compared to updating the entire model with single data.

**Updating style:** The simplest approach is computing loss after observing a single sample, and calculating gradients via backpropagation, then updating parameters using these gradients. However, as noted before, gradients estimated from just one sample are unstable. Many existing methods refine this basic scheme. A fundamental improvement is experience replay: during each loss computation, previously seen samples are also sampled and included (Lau et al., 2025). This method aggregates multiple samples for gradient calculation and stabilizes training by reducing gradient estimation variance, though unbiased estimates may be compromised. Beyond experience replay, methods like FSNet (Pham et al., 2023) propose a dual-stream network. Specifically, one stream performs fast updates using gradients only from recent samples. The other stream updates slowly, utilizing gradients aggregated over numerous past samples. Finally, instead of gradient-based updates, some approaches (e.g., ELF (Lee et al., 2025)) propose directly inferring optimal parameters. For instance, the adapter in ELF is designed as a linear model mapping inputs to outputs. Given its linear nature, the optimal parameters can be computed directly using a small batch of samples, bypassing gradient updates.

We have collected recently proposed methods for online time series forecasting and summarized them in Table 1 below based on which parameters to update and how to update them.

Beyond these approaches, some research has proposed using specific normalization techniques to address distribution shift. For instance, recognizing that the mean and variance of samples often fluctuate over time, certain studies suggest calculating normalization parameters individually for each sample (Kim et al., 2021). Others forecast future statistics based on statistics from past data (Fan et al., 2023; Liu et al., 2023c) and use more informative statistics (Ye et al., 2024; Dai et al., 2024), aiming to apply appropriate normalization so that the data input to the model at each time step follows a more standardized distribution.

It is important to emphasize that these normalization strategies primarily target covariate shift – a scenario where the relationship from $x$ to $y$ (from input to target) remains unchanged, but the distribution of the input $x$ changes. However, a key limitation arises: these methods are ineffective against concept drift – situations where the relationship from $x$ to $y$ changes.

Table 1: The summary of existing online time series prediction methods

| Method | Parameters for updating | Updating style |
|---|---|---|
| Onenet (Wen et al., 2023) | Weights of different model | Exponentiated Gradient Descent and offline reinforcement learning |
| FSNet (Pham et al., 2023) | Weights in convolutional layer and feature modification weights after convolutional layer | Gradient decent with two-stream EMA |
| D3A (Zhang et al., 2024) | Full parameters of the model | 1. Update only when distribution drift is detected. 2. Use a large amount of accumulated data to update the model and simultaneously add noise to the data |
| DSOF (Lau et al., 2025) | Parameters in an adapter mapping the output of original model and input to prediction | Online gradient decent with replay buffer |
| Proceed (Zhao & Shen, 2025) | Low-rank scaling coefficients $\alpha, \beta$ for model weights | Proactive lightweight adaptation: 1. Concept drift-triggered model update. 2. Rescaling model weight using $\alpha$ out product $\beta$ |
| SOLID (Chen et al., 2024b) | Parameters in Prediction Layer | 1. Update only when distribution drift is detected. 2. using samples with temporal proximity and periodic phase similarity input similarity to finetune |
| ELF (Lee et al., 2025) | 1. The parameter in a linear adapter mapping input to prediction 2. The emsamble weight between base model and adapter | Directly fit for parameters in adapter; Exponentiated Gradient Descent for ensamble weights |
| ADCSD (Guo et al., 2024) | The parameters in an adapter mapping the output of base model to prediction | Online gradient decent |

## 3 METHOD

### 3.1 KEY IDEA

First, we decompose a prediction model into two components: an encoder that extracts features from observed data, and a prediction head. This decomposition is common in the field of deep learning and has been employed in previous works (Teng et al., 2023; Chen et al., 2024a). We denote the encoder as $f$ and the prediction head as $g$. At time $t$, input data $x_t$ is processed by the encoder $f$ and the feature representation $z_t$ can be obtained and the prediction head use $z_t$ to output the prediction for future time steps. Besides, the true target is denoted as $y_t$ and due to potential model obsolescence, $g(z_t)$ is not equal to $y_t$. Our objective is therefore to find a correction term, denoted $\delta_t$, such that $g(z_t + \delta_t)$ approximates $y_t$.

### 3.2 LEARNING ADJUST TERM USING ADAPTER

The most straightforward approach to compute $\delta_t$ is conducting gradient descent to the feature representation. For example, if we conduct $k$-step ahead forecast, this procedure can be expressed as follows:

$$\delta_{t+1} = \delta_t - \eta \frac{\partial (g(z_{t-k} + \delta_{t-k}) - y_{t-k})^2}{\partial \delta_{t-k}} \tag{1}$$

Where $\eta$ is learning rate. While this straightforward approach could lead to some improvement, its effectiveness remains limited in our experiments. We attribute this to two core issues. First, multi-step prediction creates a delay: at time $t$, we cannot observe the true value $y_t$ for a $k$-step ahead forecast. Only the pair $(x_{t-k}, y_{t-k})$ is available to compute the loss and update $\delta_t$, which could introduce potentially harmful lag. Second, the optimal correction $\delta_t$ might not be a constant and could depend on background conditions. As a result, the required correction $\delta_t$ might be correlated with the state represented by $z_t$.

To address these limitations, we propose to use a small neural network that maps the current feature vector $z_t$ to the correction $\delta_t$. This small neural network can be named as "adapter". This adapter, typically a simple MLP, can be calibrated using a held-out validation set. Moreover, during deployment, as true data arrives sequentially, we can continuously update the adapter's parameters online via gradient descent. This solution offers significant advantages: 1) it can output different corrections based on the contextual features $z_t$, and 2) since $\delta_t$ is predicted directly from $z_t$ (available immediately at time $t$), it sidesteps the multi-step prediction delay problem.

Besides, past gradients of features also contain valuable information. Therefore, our final adapter architecture takes two inputs: the current feature vector $z_t$ and the historical gradients calculated from previous steps. These inputs are fused within the adapter network to predict $\delta_t$.

Moreover, directly concatenating or summing $z_t$ and the gradients as input to a single MLP proves problematic because the magnitude of gradient is often much smaller than that of features. To handle this disparity, our adapter uses a dual-path structure: separate linear layers first transform the features $z_t$ and the historical gradients independently. Their outputs are then summed, and this combined representation passes through another two linear layers to produce the final correction $\delta_t$.

### 3.3 Computing Historical Gradients for Adapter Input

As noted earlier, using a single sample to calculate gradients often results in high variance. To mitigate this, we employ a batch-based approach. Specifically, given a batch size $b$ a prediction horizon of $k$ steps, at time $t$, we compute the average loss using the predictions and truth values from time stamps $t - k - b$ to $t - k$. The gradients derived from this averaged loss are then used as the historical gradient input to the adapter. This batch-based calculation can reduce gradient variance compared to the single-sample estimation.

### 3.4 Online update

During deployment, we also continuously update the adapter's parameters using online gradient descent. However, due to the delay inherent in multi-step forecasting, we implement a $k$-step delayed online gradient descent method. Specifically, we cache the historical gradients, features and the corresponding model outputs for each timestep. Upon observing the true value at time $t$, we calculate the loss associated with the prediction made at $t - k$. This loss is then used to propagate gradients backward and update the adapter's parameters. Concurrently, we also perform online updates of the parameters in the model's final linear layer during deployment.

## 4 Experiments

### 4.1 Setup

We selected 13 commonly used datasets for time series forecasting, including four ETT datasets (ETTh1, ETTh2, ETTm1, ETTm2), four PEMS datasets (PEMS03, PEMS04, PEMS07, PEMS08), and five additional datasets: weather, solar, traffic, electricity, and exchange. All datasets originate from the iTransformer paper (Liu et al., 2024). To demonstrate the broad applicability of our proposed method across various forecasting models, we selected three prediction models: iTransformer (Liu et al., 2024), SOFTS (Han et al., 2024), and TimesNet (Wu et al., 2023). The architectures of these models can be decomposed into sequential blocks. We regard the output from the second last block as the feature representation.

For the forecasting task, we adopted the same setup as the DSOF paper (Lau et al., 2025): models take the past 96 time steps as inputs and predict the next 12, 24, and 48 steps.

Regarding dataset splits, our approach differs from prior work. Earlier studies typically allocated the first 25% of data for training, the middle 5% for validation, and the final 70% for testing (Lau et al., 2025; Zhao & Shen, 2025), primarily to ensure an extended online deployment phase. However, this split is probably unrealistic. For example, in the ETT datasets, this would create a deployment period spanning nearly two years – an long interval during which the model is highly likely to be retrained. Instead, we employed a more standard chronological partitioning: 60% for training, 10%

for validation, and 30% for testing. The code of our method is available at `https://github.com/xiannanhuang/ADAPT-Z`.

## 4.2 BASELINE METHODS

We evaluated four state-of-the-art time series online learning methods: DSOF (Lau et al., 2025), Proceed (Zhao & Shen, 2025), ADCSD (Guo et al., 2024), and SOLID (Chen et al., 2024b). Additionally, we included two fundamental baselines: traditional (delayed) online gradient descent (applied to all model parameters) and feature-space online gradient descent (only updating the feature representations, we name this method as fOGD). Two other online methods—OneNet (Wen et al., 2023) and FSNet (Pham et al., 2023)—were considered too. But it should be noted that they require proprietary forecasting model architectures rather than functioning as plug-in solutions for arbitrary point predictors, which differs from other methods.

## 4.3 RESULTS

### 4.3.1 MAIN RESULTS

Table 2: Main forecast results. The results in this table show the **average MSE** for three base models and three prediction steps. More detailed results are shown in the Appendix. The last column shows the percentage error reduction of our method compared to the original models. The best result is colored in **red** and the second is colored in **blue** with an underline.

| Method | Ori | fOGD | OGD | DSOF | SOLID | ADCSD | Proceed | FSNET | OneNet | ADAPT-Z | IMP |
|---|---|---|---|---|---|---|---|---|---|---|---|
| ETTh1 | 0.2782 | 0.2771 | 0.2777 | 0.3220 | 0.2764 | 0.2777 | 0.2766 | 1.0220 | 0.7806 | 0.2657 | 4.47% |
| ETTh2 | 0.1648 | 0.1648 | 0.1649 | 0.1926 | 0.1645 | 0.1648 | 0.1645 | 1.1110 | 0.9996 | 0.1604 | 2.72% |
| ETTm1 | 0.2211 | 0.2178 | 0.2180 | 0.2647 | 0.2166 | 0.2169 | 0.2168 | 0.9806 | 1.0079 | 0.1937 | 12.42% |
| ETTm2 | 0.0973 | 0.0971 | 0.0972 | 0.1405 | 0.0974 | 0.0970 | 0.0974 | 1.0559 | 0.9543 | 0.0937 | 3.67% |
| PEMS03 | 0.0987 | 0.0975 | 0.0991 | 0.1645 | 0.1003 | 0.0982 | 0.1003 | 0.4106 | 0.4093 | 0.0959 | 2.86% |
| PEMS04 | 0.1288 | 0.1263 | 0.1285 | 0.1465 | 0.1291 | 0.1280 | 0.1290 | 0.4745 | 0.4729 | 0.1223 | 5.05% |
| PEMS07 | 0.0920 | 0.0908 | 0.0921 | 0.1113 | 0.0919 | 0.0909 | 0.0920 | 0.4860 | 0.4851 | 0.0892 | 3.04% |
| PEMS08 | 0.1498 | 0.1482 | 0.1497 | 0.1790 | 0.1506 | 0.1488 | 0.1508 | 0.6086 | 0.6085 | 0.1426 | 4.80% |
| electricity | 0.1138 | 0.1126 | 0.1136 | 0.1119 | 0.1136 | 0.1132 | 0.1136 | 0.7802 | 0.7806 | 0.1096 | 3.68% |
| exchange | 0.0433 | 0.0432 | 0.0433 | 0.0493 | 0.0432 | 0.0432 | 0.0432 | 2.3596 | 1.2305 | 0.0429 | 0.98% |
| solar | 0.1084 | 0.1074 | 0.1072 | 0.1038 | 0.1083 | 0.1075 | 0.1083 | 0.1124 | 0.1966 | 0.0948 | 12.61% |
| traffic | 0.4075 | 0.4068 | 0.4033 | 0.4060 | 0.4070 | 0.4070 | 0.4079 | 0.8713 | 0.5003 | 0.3689 | 9.49% |
| weather | 0.1575 | 0.1573 | 0.1578 | 0.1975 | 0.1573 | 0.1564 | 0.1575 | 0.6116 | 0.5490 | 0.1481 | 5.98% |

Table 2 presents forecast results measured by average Mean Squared Error (MSE) across 13 datasets. It can be observed that Adapter-Z consistently achieves the best performance on every dataset.

The final column (IMP) quantifies Adapter-Z's improvement by showing the percentage reduction in error compared to the original prediction models for each dataset. These improvements ranging from significant reductions exceeding 5% (e.g., 12.42% on ETTm1, 12.61% on solar, and 9.49% on traffic) to moderate improvements between 2-5% (e.g., 4.47% on ETTh1, 5.05% on PEMS04) and smaller gains like the 0.98% on exchange.

Furthermore, it can be observed that applying online gradient descent at the feature level achieves the second-best results on many datasets. This suggests that modifying features is likely a promising direction for online time series forecasting. Even a relatively simple approach like OGD demonstrates meaningful accuracy improvements.

### 4.3.2 FINETUNE USING TRAINING SET

It is notable that approaches like DSOF, Onenet and FSNet utilize the training set during their process. Therefore, we also propose two enhanced versions of our method that using the training set. Specifically, we finetune the base model and train the adapter on the training set for 3 epochs before online deployment. And we considered two distinct deployment versions: 1) Version 1: During online deployment, we continue to perform online fine-tuning of the adapter. 2) Version 2: During online deployment, all parameters of the base model and adapter are frozen (i.e., no online updating.). We tested this variant based on the iTransformer architecture.

The results in Table 3 show that for most datasets, using the training data to finetune the base model and initialize the adapter helps reduce prediction error. This makes sense because in our earlier setup, we only used the validation data - so the adapter was likely not fully trained. Moreover, the results

Table 3: The results of experiments using the training set

| Dataset | ETTh1 | ETTh2 | ETTm1 | ETTm2 | PEMS03 | PEMS04 | PEMS07 | PEMS08 | electricity | exchange | solar | traffic | weather |
|---|---|---|---|---|---|---|---|---|---|---|---|---|---|
| Ori | 0.2756 | 0.1635 | 0.2309 | 0.0970 | 0.1018 | 0.1397 | 0.0932 | 0.1578 | 0.1015 | 0.0394 | 0.1237 | 0.3550 | 0.1550 |
| fOGD | 0.2730 | 0.1632 | 0.2233 | 0.0967 | 0.0987 | 0.1343 | 0.0896 | 0.1546 | 0.0992 | 0.0392 | 0.1189 | 0.3512 | 0.1531 |
| ADAPT-Z | **0.2626** | **0.1582** | **0.1954** | **0.0941** | 0.0974 | 0.1264 | 0.0892 | 0.1453 | 0.0971 | 0.0384 | **0.0940** | **0.3314** | **0.1461** |
| Version1 | **0.2625** | **0.1573** | **0.1948** | **0.0940** | **0.0936** | **0.1192** | **0.0865** | **0.1342** | **0.0939** | **0.0379** | **0.0885** | **0.3197** | **0.1455** |
| Verson2 | 0.2680 | 0.1598 | 0.2104 | 0.0963 | **0.0945** | **0.1196** | **0.0877** | **0.1351** | **0.0959** | **0.0378** | 0.1141 | 0.3224 | 0.1490 |

of Version 2 show that the test error can be reduced even without updating any parameters during online deployment, and the error is usually smaller than that of conducting online gradient descent on the feature space (fOGD). This means the model actually learned how to learn. Because during training, the model can see both the data in current batch and the error from the previous batch. As a result, it could learn to adjust predictions in changing environments by using information from the previous batch even without parameter changing. This reveals a key gap in current training process of time series prediction model: it always treats training samples as independent, shuffling their order randomly in training. But in real online deployment, samples arrive in order, and models can use past information to adjust future predictions. In other word, the training style and test style is mismatch. Therefore, future work could consider sample order during training to fill this gap and our experiment here is a small step toward that.

### 4.3.3 ANALYSIS OF FEATURE LOCATION

In our main experiments, since all three point forecasting models can be divided into several blocks, we used the output from the second last block as features. As a result, this raise two questions: whether this specific block selection optimal and what would happen if using features from other locations?

To investigate this, we conducted additional experiments using the iTransformer model across some datasets. Figure 2 shows the architecture of iTransformer model: starting with the input layer, followed by RevIN normalization, an embedding layer that projects time series dimensions to d_model, three Transformer blocks, a projection layer mapping back to time series dimensions, and finally RevIN denormalization. We tested online prediction performance using outputs from each layer as features. Table 4 presents the results: baseline without online tuning (Row 1), regarding raw input as features (Row 2), regarding RevIN layer outputs as features(Row 3), regarding embedding layer outputs (Row 4) as features, etc.

Table 4 reveals that different datasets favor distinct feature locations. For the electricity dataset, optimal features occur between the projection layer and the final denormalization layer. For solar data, features from the first transformer block result in superior results. Despite these variations, the performance of our method remains stable when selecting the output of different intermediate layers as features. However, directly adjusting the input data consistently performs poorly and often results in worse results than the baseline model without adaptation. When averaging the performance across all five datasets, we ultimately found the output after the first transformer block delivered the optimal results.

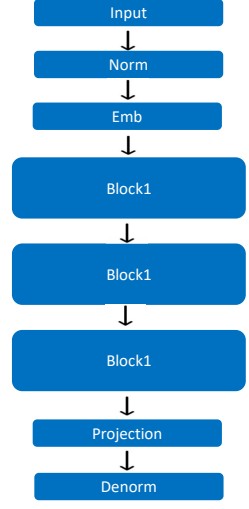

Figure 2: The structure of iTransformer

To further investigate why certain feature layers are more suitable for our method, we computed the Root Mean Square Difference (RMSD) of feature gradients across time steps, which measures gradient stability. Additionally, we calculated the Mean Absolute Value (MAV) of feature gradients to reflect the magnitude of distribution shift. We then examined whether these statistics correlate with ADAPT-Z's performance. The results are presented in Table 5. We

Table 4: The results of online prediction experiments with different feature locations

| Dataset | electricity | | | PEMS03 | | | PEMS07 | | | solar | | | weather | | | |
|---|---|---|---|---|---|---|---|---|---|---|---|---|---|---|---|---|
| H | 1 | 24 | 48 | 1 | 24 | 48 | 1 | 24 | 48 | 1 | 24 | 48 | 1 | 24 | 48 | Mean |
| ori | 0.0534 | 0.1126 | 0.1386 | 0.0388 | 0.0974 | 0.1691 | 0.0408 | 0.0938 | 0.1450 | 0.0087 | 0.1231 | 0.2393 | 0.0547 | 0.1878 | 0.2225 | 0.1150 |
| Input | 0.0657 | 0.1407 | 0.2788 | 0.0928 | 0.1050 | 0.1841 | 0.0430 | 0.0955 | 0.1503 | 0.0082 | 0.1196 | 0.1894 | 0.0541 | 0.2104 | 0.2343 | 0.1315 |
| Norm | 0.0495 | 0.1085 | 0.1326 | 0.0383 | 0.0922 | 0.1572 | 0.0391 | 0.0904 | 0.1363 | 0.0080 | 0.1062 | 0.1988 | 0.0537 | 0.1783 | 0.2127 | 0.1068 |
| Emb | 0.0484 | 0.1089 | 0.1331 | 0.0380 | 0.0914 | 0.1615 | 0.0389 | 0.0897 | 0.1385 | 0.0077 | 0.0964 | 0.1750 | 0.0503 | 0.1702 | 0.2157 | 0.1043 |
| Block 1 | 0.0483 | 0.1087 | 0.1331 | 0.0380 | 0.0915 | 0.1610 | 0.0389 | 0.0894 | 0.1378 | 0.0077 | 0.0971 | 0.1750 | 0.0503 | 0.1699 | 0.2144 | 0.1041 |
| Block 2 | 0.0484 | 0.1089 | 0.1339 | 0.0381 | 0.0922 | 0.1617 | 0.0389 | 0.0899 | 0.1386 | 0.0077 | 0.0972 | 0.1771 | 0.0505 | 0.1730 | 0.2147 | 0.1047 |
| Block 3 | 0.0487 | 0.1094 | 0.1346 | 0.0382 | 0.0940 | 0.1650 | 0.0391 | 0.0906 | 0.1414 | 0.0077 | 0.1013 | 0.1849 | 0.0501 | 0.1791 | 0.2166 | 0.1067 |
| Projection | 0.0494 | 0.1065 | 0.1297 | 0.0384 | 0.0945 | 0.1560 | 0.0390 | 0.0874 | 0.1389 | 0.0080 | 0.1105 | 0.2035 | 0.0535 | 0.1793 | 0.2106 | 0.1070 |
| Denorm | 0.0510 | 0.1118 | 0.1379 | 0.0477 | 0.0919 | 0.1640 | 0.0390 | 0.0874 | 0.1275 | 0.0080 | 0.1101 | 0.1774 | 0.0525 | 0.1745 | 0.2109 | 0.1061 |

Table 5: The relationship between gradient statistics and prediction errors

| Dataset | H | 1 | 24 | 48 |
|---|---|---|---|---|
| electricity | MAV | 0.1243 | 0.2482 | 0.1239 |
| | RMSD | 0.5432 | 0.4119 | 0.4790 |
| PEMS03 | MAV | -0.3240 | 0.2934 | 0.1528 |
| | RMSD | 0.5382 | 0.6253 | 0.4921 |
| PEMS07 | MAV | 0.1943 | -0.0835 | 0.0832 |
| | RMSD | 0.6341 | 0.5612 | 0.6267 |
| solar | MAV | 0.1432 | 0.2732 | 0.3854 |
| | RMSD | 0.6390 | 0.5192 | 0.5715 |
| weather | MAV | 0.0211 | -0.0982 | 0.1693 |
| | RMSD | 0.4836 | 0.4293 | 0.6923 |

observe that RMSD exhibits a clear positive correlation with final MSE, suggesting that more stable gradients (lower RMSD) during deployment lead to better prediction performance. In other words, feature representations with consistent gradient directions are more beneficial for online adaptation than those with highly volatile gradients.

### 4.3.4 ABLATION EXPERIMENTS

We also conducted ablation studies for our method. Since our method takes two inputs—historical gradients and current features—we designed two ablation experiments: First, we removed the historical gradient input (w\o grad). Second, we removed the current feature input (w\o feature). We conducted these experiments using SOFTS model across all datasets. Table 6 shows the results. It can be observed that both ablated versions showed higher MSE than the original ADAPT-Z method. This proves that keeping both historical gradients and current features is necessary.

Moreover, both ablated models still outperformed the basic prediction model. This confirms that both current features and historical gradients provide useful information for online prediction.

### 4.3.5 DISCUSSION ON FINETUNING EPOCH NUMBER

Our method involves finetuning an adapter for three epochs on the validation set before online deployment, which naturally raises two questions: whether three epochs represents the optimal tuning duration, and how performance would be in extreme scenarios where validation dataset is not obtainable. To answer these questions, we tested different tuning epochs: zero epoch, one epoch, three epochs, five epochs and ten epochs. we use SOFTS model as basic prediction model.

Figure 3 shows that the best training epoch varies across datasets. For example, five epochs work best for the ETTh1 dataset, while more epochs may be required for the PEMS03 dataset. However, our method still achieves good accuracy even without validation set training. It outperforms the basic model without online prediction. Also, it exceeds the best baseline online prediction method on most datasets.

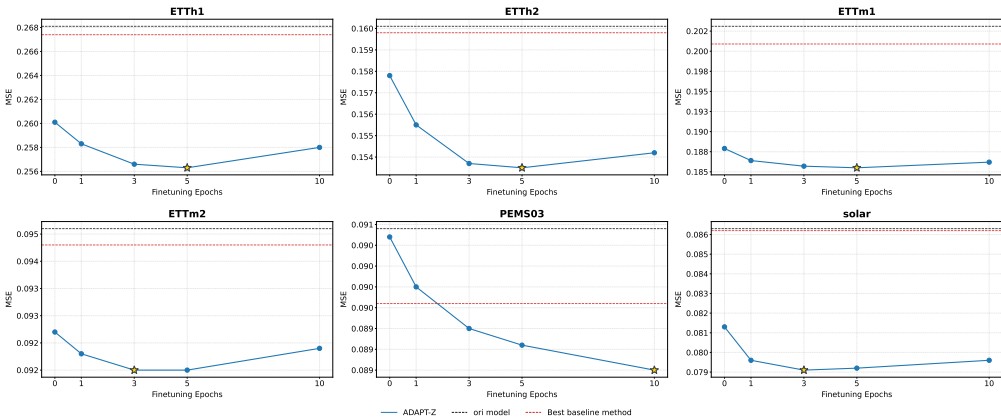

Figure 3: The relationship between the number of training epochs on the validation set and the final MSE during deployment. The yellow star marks the epoch number that achieved the optimal MSE. The black dashed line shows the MSE when using the original model directly on the test set. The red dashed line represents the MSE of the best baseline online deployment method. The MSE is the average result of three prediction horizons.

Table 6: The results of ablation experiments

|  | Ori | ADAPT-Z | w\o grad | w\o feature |
|---|---|---|---|---|
| ETTh1 | 0.2681 | 0.2566 | 0.2579 | 0.2611 |
| ETTh2 | 0.1601 | 0.1537 | 0.1554 | 0.1577 |
| ETTm1 | 0.2031 | 0.1857 | 0.1862 | 0.1919 |
| ETTm2 | 0.0946 | 0.0920 | 0.0921 | 0.0928 |
| PEMS03 | 0.0907 | 0.0895 | 0.0900 | 0.0906 |
| PEMS04 | 0.1236 | 0.1196 | 0.1201 | 0.1211 |
| PEMS07 | 0.0878 | 0.0865 | 0.0870 | 0.0875 |
| PEMS08 | 0.1129 | 0.1094 | 0.1100 | 0.1121 |
| electricity | 0.0976 | 0.0961 | 0.0965 | 0.0969 |
| exchange | 0.0376 | 0.0371 | 0.0370 | 0.0373 |
| solar | 0.0862 | 0.0791 | 0.0794 | 0.0839 |
| traffic | 0.3149 | 0.3109 | 0.3119 | 0.3138 |
| weather | 0.1571 | 0.1457 | 0.1506 | 0.1514 |

Table 7: The results of experiments combining normalization-based methods and our approach. Dish+ and FAN+ means the experiments combining Dish or FAN with ADAPT-Z together

|  | Ori | DishTS | DishTS+ | FAN | FAN+ | ADAPT-Z |
|---|---|---|---|---|---|---|
| ETTh1 | 0.2756 | 0.2928 | 0.2666 | 0.2709 | 0.2657 | 0.2626 |
| ETTh2 | 0.1635 | 0.1709 | 0.1607 | 0.1611 | 0.1581 | 0.1582 |
| ETTm1 | 0.2309 | 0.2205 | 0.1914 | 0.2061 | 0.2037 | 0.1954 |
| ETTm2 | 0.0970 | 0.1119 | 0.0967 | 0.0973 | 0.0960 | 0.0941 |
| PEMS03 | 0.1018 | 0.0987 | 0.0898 | 0.1126 | 0.1094 | 0.0974 |
| PEMS04 | 0.1397 | 0.0943 | 0.0884 | 0.1317 | 0.1233 | 0.1264 |
| PEMS07 | 0.0932 | 0.0934 | 0.0850 | 0.1168 | 0.1116 | 0.0892 |
| PEMS08 | 0.1578 | 0.1489 | 0.1304 | 0.1564 | 0.1527 | 0.1453 |
| electricity | 0.1015 | 0.0982 | 0.0925 | 0.1093 | 0.1045 | 0.0971 |
| exchange | 0.0394 | 0.0792 | 0.0524 | 0.0485 | 0.0433 | 0.0384 |
| solar | 0.1237 | 0.0930 | 0.0776 | 0.1112 | 0.0913 | 0.0940 |
| traffic | 0.3550 | 0.3910 | 0.3488 | 0.4108 | 0.4076 | 0.3314 |
| weather | 0.1550 | 0.1515 | 0.1408 | 0.1486 | 0.1450 | 0.1461 |

### 4.3.6 COMPATIBILITY WITH NORMALIZATION BASED METHOD

Some normalization-based methods like DishTS (Fan et al., 2023) and FAN (Ye et al., 2024) address distribution shift in time series prediction by dynamically adjusting normalization parameters per sample to stabilize input distributions. These methods could complement ours, and combining them with our method may yield better results since we address online prediction from different perspectives. To validate this hypothesis, we conducted experiments comparing standalone deployments of DishTS and FAN against methods integrating each normalization technique with our ADAPT-Z framework (denoted as Dish+ and FAN+). These experiments are based on iTransformer model and the results are presented in Table 7.

It can be observed that when combining normalization methods with our online adaptation approach, the results consistently surpass standalone normalization performance. This is evident when comparing the "DishTS" versus "DishTS+" columns and "FAN" versus "FAN+" columns in Table 7. This demonstrates that our feature-space adaptation mechanism effectively amplifies the benefits of advanced normalization techniques like DishTS and FAN.

Contrary to findings in original normalization papers, additional normalization doesn't universally improve performance across all datasets. This occurs for two primary reasons: First, our baseline

already incorporates RevIN – a lightweight adaptive normalization method – creating diminishing returns for extra normalization layers. Second, the original paper conducted longer-horizon forecasts (e.g., predicting 96 or even 192 future steps), our shorter prediction horizons (1-48 steps) experience more volatile distribution shifts. This instability makes it challenging for normalization method to consistently align input-output distributions across diverse time segments.

Finally, in datasets where normalization demonstrates effectiveness, combining normalization with ADAPT-Z leads to the best results. For example, on the ETTm1 dataset, using DishTS alone reduced MSE from 0.2309 to 0.2037 and combining DishTS with ADAPT-Z drove MSE down to 0.1914 – surpassing all other methods.

## 5  LIMITATIONS

While ADAPT-Z demonstrates consistent improvements across multiple datasets and models, we acknowledge several limitations of our work: **Feature Layer Selection.** Our analysis shows that optimal feature extraction layers vary across datasets and models. While we identify a correlation between gradient stability (RMSD) and prediction performance, we do not provide a principled method for automatically selecting the best feature layer without empirical tuning. Developing automated layer selection strategies remains an important direction for future research. **Computational Overhead.** Although ADAPT-Z is memory-efficient, it introduces additional runtime compared to some baseline methods (e.g., simple normalization-based approaches). For ultra-low-latency applications requiring sub-second predictions, this trade-off between accuracy and speed should be carefully considered. **Generalization to Linear Models.** ADAPT-Z assumes the forecasting model can be decomposed into an encoder and prediction head, which enables feature-space adaptation. This design limits direct application to simple linear models that lack such modular structure. Extending feature-space adaptation to linear architectures would broaden the method's applicability.

## 6  CONCLUSION

In summary, we propose **ADAPT-Z**, a novel paradigm for online time series forecasting by addressing distribution shifts through feature-space correction. This method integrates current contextual features and historical gradient information, effectively resolves the critical challenge of delayed feedback in multi-step predictions while ensuring update stability. Experiments across 13 diverse datasets and 3 base models demonstrate that adjusting feature representations of underlying factors—rather than conventional model parameters—enables more effective adaptation to non-stationary environments. Certainly, our work is not without limitations, and several promising directions remain for future exploration such as developing more effective sample selection strategies for gradient computation (e.g., through experience replay or hard sample replay), designing enhanced adapter architectures. Additionally, exploring methods to properly utilize sample ordering during the training phase represents another key direction for future research.

## REPRODUCIBILITY STATEMENT

We have made every effort to ensure that the results presented in this paper are reproducible. All code and datasets have been made publicly available in an anonymous repository to facilitate replication and verification. The experimental setup, including training steps, model configurations, and hardware details, is described in detail in the paper. Additionally, the datasets we used are publicly available, ensuring consistent and reproducible evaluation results. We believe these measures will enable other researchers to reproduce our work and further advance the field.

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

# A MORE RELATED WORK

## A.1 CONTINUAL LEARNING

Continual learning tackles the challenge of learning new patterns in a changing environment without catastrophic forgetting (Wang et al., 2024). The main approaches include regularization-based methods, which add constraints to preserve important parameters for old tasks (Ahn et al., 2019). For example, Elastic Weight Consolidation (EWC) penalizes changes to critical weights using Fisher information matrices (Kirkpatrick et al., 2017). Besides, replay-based methods store or regenerate past data for retraining: experience replay keeps real samples (Castro et al., 2018), while generative replay uses models like GANs to create synthetic examples (Liu et al., 2022). These techniques help maintain stability but face storage or data quality challenges. Other strategies involve dynamic architecture methods that expand models for new tasks—such as adding task-specific branches or adapter modules—to isolate parameters and prevent interference (Ramesh & Chaudhari, 2022; Ostapenko et al., 2021). Moreover, optimization based approaches like Gradient Episodic Memory (GEM) adjust optimization directions to balance old and new task learning (Lopez-Paz & Ranzato, 2017).

Overall, there are many methods in the current field of online time series prediction that are similar to continuous learning methods, such as methods based on experience replay (Lau et al., 2025) or finding key parameters (Wen et al., 2023).

## A.2 ONLINE CONVEX OPTIMIZATION

Though our problem is not strictly an Online Convex Optimization (OCO) problem—as the loss for neural network parameters is typically complex and non-convex, theoretical OCO work still provides valuable inspiration. Online optimization involves an algorithm (the "player") that makes a decision at each step, observes an incurred loss, and uses that loss iteratively to update subsequent decisions. The core objective is to minimize regret, defined as the cumulative difference in utility between the algorithm's decisions and the best decision chosen in hindsight (Hazan, 2016). In online time series forecasting, we define utility as the negative loss at each step. This aligns with dynamic regret, which compares performance against a potentially changing optimal decision sequence over time.

Recent theoretical work has established bounds on dynamic regret. A key classic upper bound shows that for an algorithm using OGD for the true loss at each step, the regret bound depends on the $l_2$-norm of the cumulative difference between optimal parameters across the prediction horizon (Zinkevich, 2003; Mokhtari et al., 2016). However, deep learning optimization is inherently more complex: the true target is an expected loss, but we can only compute losses on observed samples, introducing gradient estimation error. Consequently, research shows that applying OGD under such gradient noise yields a regret bound incorporating an additional term related to the variance of these gradient estimates (Yang et al., 2016).

## A.3 DOMAIN GENERALIZATION

Domain Generalization (DG) is formally defined as training models on data from multiple source domains to achieve robust performance on unseen target domains with different distributions. While DG shares similarities with online prediction in handling train-test distribution shifts, DG focuses more on training strategies that inherently improve out-of-domain robustness and online prediction focuses on using pre-trained models in dynamic environments.

Existing Domain Generalization approaches primarily fall into three categories. **Data Manipulation** enhances diversity through input transformations like style randomization (Tobin et al., 2017) or generate novel domain samples using generative models to broaden training coverage (Qiao et al., 2020). **Representation Learning** employs adversarial training or kernel methods to align feature distributions across domains for domain-invariant representations (Liu et al., 2018; Ganin et al., 2015). Complementing these, **Learning Strategy Optimization** methods include meta-learning frameworks that simulate domain shifts during training to foster adaptation capabilities (Dou et al., 2019), gradient-based approaches that directly regularize gradient behaviors for stable cross-domain generalization(Nasery et al., 2021), and ensemble techniques that combine domain-specific experts for collective decision-making (Mancini et al., 2018).

### A.4 Test time adaption

Test-time adaptation (TTA) refers to techniques that dynamically adjust pre-trained models using unlabeled test data during the deployment phase to improve performance on target data distributions. Initially, TTA primarily focused on classification tasks and can be broadly categorized into optimization-based, data-based, and model-based approaches. Optimization-based methods typically involve recalibrating normalization layer statistics (e.g., in BatchNorm), designing unsupervised loss functions like entropy minimization (Wang et al., 2021), or utilizing a mean-teacher framework for stable updates (Wang et al., 2022). Data-based strategies focus on diversifying test batches via data augmentation (Zhang et al., 2022) or preserving valuable information using memory banks for replay (Yuan et al., 2023). Model-based techniques adapt the architecture itself, such as adding input transformation or adapter modules (Liu et al., 2023b) or incorporating prompt-tuning layers, particularly in vision-language models (Liu et al., 2023a). While prevalent in classification, these methods face challenges in time series forecasting due to architectural differences (e.g., lack of standard normalization layers) and the inapplicability of common image-centric augmentations to temporal data.

In recent years, several works have explored TTA for time series, such as ADCSD (Guo et al., 2024), PETSA (Medeiros et al., 2025), and TAFAS (Kim et al., 2025). These methods can be summarized based on two aspects introduced earlier: which parameters to update and how to update them. In terms of parameter selection, all these methods employ adapters, but they differ in structure and application. For example, ADCSD applies the adapter after the model output, while PETSA and TAFAS apply adapters to both the model input and output. Additionally, PETSA incorporates a LoRA-like structure in its adapter. Regarding update strategies, ADCSD uses traditional gradient descent, while other methods propose training with partially observed data.

## B  Implementation details

**Pseudo code:**

---

**Algorithm 1** Automatic Delta Adjustment via Persistent Tracking in Z-space

---

**Require:** A prediction model, which includes an encoder $f$ and a prediction head $g$, Adapter network $A$, Prediction horizon $k$, batch size $b$ for computing historical gradients;

1: initialize history gradient $hisgrad_1 = 0$
2: **for** each time step $t$ **do**
3:      Extract features: $z_t = f(x_t)$
4:      Compute adjust term: $\delta_t = A(z_t, hisgrad_{t-k})$
5:      Obtain predictions: $\hat{y}_t = g(z_t + \delta_t)$
6:      Observe $y_{t-k+1}$
7:
8:      **if** $t \geq k + b$ **then**
9:          Compute history gradient of feature as:

$$hisgrad_{t-k+1} = \frac{\left(g(z_{t-k-b+1:t-k+1}) - y_{t-k-b+1:t-k+1}\right)^2}{\partial z}$$

10:          Compute loss: $loss = MSE(\hat{y}_{t-k-b+1:t-k+1}, y_{t-k-b+1:t-k+1})$
11:          Update parameters of $A$ and the last layer of $g$ using $loss$.
12:      **else**
13:          Set $hisgrad_{t-k+1} = 0$
14:      **end if**
15: **end for**

---

**Hyperparameters:** We consistently employed a batch size of 24 when computing historical gradients throughout our experiments. Subsequent sections include sensitivity analyses examining the impact of varying batch sizes on forecasting accuracy. During validation set training of the adapter, a learning rate of 0.001 was applied, while online deployment utilized a learning rate of 0.0003 for finetuning the adapter module and 0.00003 for finetuning the model's final linear layer. This

consistent configuration was maintained across all datasets and base forecasting models. Additional sensitivity analysis regarding online fine-tuning learning rates will be presented in later sections. The detailed structure of adapter is provided in Figure 4. In the experiment, we use a batch-wise prediction style with batch size 24 rather than single sample prediction to accelerate the online process.

**Baseline methods:** For the fOGD method, we set its key parameter (learning rate) to 0.001. This value was chosen after testing multiple options and proved generally effective.

All implementations of other baseline methods came from official sources. DSOF used its publicly released code, FSNet and OneNet implementations also came from this repository. SOLID and Proceed methods were implemented using code from Proceed's open-source repository. Importantly, Proceed originally requires training data to calibrate its concept encoder, we maintained fairness by using only validation data for calibration—consistent with how we trained our method's adapter.

For base forecasting models, we used default hyperparameters (like layer dimensions and depth) provided in their official codebases or reported as optimal in papers. Since tuning base models wasn't our research focus, we didn't explore alternative configurations.

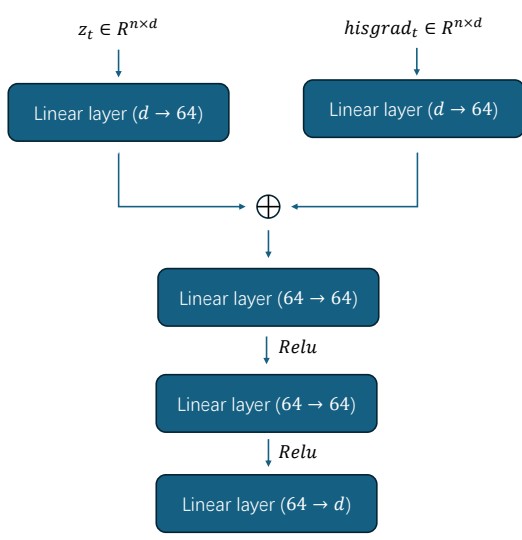

Figure 4: The structure of adapter

Regarding experimental reproducibility: our proposed method becomes fully deterministic once the base forecasting model is trained. Similarly, SOLID, ADCSD and some other baselines also exhibit no randomness after the training of base model. Consequently, all main experiments used a fixed random seed (2025) to train the base model. To demonstrate generalizability, we conducted additional replicates using seeds 2024 and 2026 for the iTransformer-based implementation; these supplementary results will appear in the following section.

## C  A CONTEXTUAL THEOREM ABOUT FINETUNING ON FEATURE SPACE

With reference to (Yang et al., 2016), we can derive an upper bound of prediction error for an online gradient descent algorithm with estimated gradient.

Suppose we at each time step $t$, $y_t$ is generated as $y_t = g_t(x_t) + \epsilon_t$, where $\epsilon_t$ represents zero-mean noise. We approximate $g_t$ using a parameterized function $f(\theta)$ and update $\theta$ at each time $t$ using gradient descent. The aim is to minimize the square loss over $T$ steps:

$$\text{loss} = \sum_{t=1}^{T} \mathbb{E}\left[(y - f(x_t|\theta_t))^2\right]. \tag{2}$$

We update $\theta$ using OGD:

$$\theta_{t+1} = \theta_t - \gamma \times \text{grad}_t, \tag{3}$$

where $\text{grad}_t$ is the true gradient with respect to $\theta_t$:

$$\text{grad}_t = \nabla_{\theta_t} \mathbb{E}\left[(y_t - f(x_t|\theta_t))^2\right]. \tag{4}$$

However, since we can only approximate the expectation using samples, we use sample-wise OGD:

$$\theta_{t+1} = \theta_t - \gamma \times \eta_t, \tag{5}$$

where $\eta_t$ is a random vector computed using the sample-wise loss. Defining the optimal parameter at time $t$ as:

$$\theta_t^* = \arg\min_{\theta_t} \mathbb{E}\left[(f(x_t|\theta_t) - y_t)^2\right], \tag{6}$$

the dynamic regret is defined as:

$$R_d = \sum_{t=1}^{T} \mathbb{E}\left[(f(x_t|\theta_t^*) - f(x_t|\theta_t))^2\right]. \tag{7}$$

Let

$$V = \sum_{t=1}^{T-1} \|\theta_t^* - \theta_{t+1}^*\|_2 \tag{8}$$

denote the path variation of optimal solutions, and $b = \max_t E\|\text{grad}_t - \eta_t\|$ bound the gradient estimation bias, and $\lambda = \max_t \text{tr}(\Sigma_{\eta_t})$ bound the trace of the covariance matrix of the estimated gradient.

**Theorem C.1** *Under the following assumptions:*

1. *$f$ is convex of $\theta$,*

2. *$\|\theta\| \leq r$ for all $\theta$,*

3. *$\|\text{grad}\|^2 \leq G$ and $E\|\eta_t\|^2 \leq G$ for all gradients.*

4. *$\nabla_{\theta_t} \mathbb{E}\left[(y_t - f(x_t|\theta_t^*))^2\right] = 0$*

*the dynamic regret is bounded by:*

$$R_d \leq Trb^2 + \frac{r}{\gamma}V + \frac{T\gamma(G+\lambda)}{2}. \tag{9}$$

Consequently, the upper bound on the expected deployment loss is:

$$\mathbb{E}\sum_{t=1}^{T}(f(x_t|\theta_t) - y_t)^2 \leq T\left(rb^2 + \frac{\gamma(G+\lambda)}{2}\right) + \frac{r}{\gamma}V + \sum_{t=1}^{T}\mathbb{E}\left[(f(x_t|\theta_t^*) - g(x_t))^2\right] + \sum_{t=1}^{T}\epsilon_t^2. \tag{10}$$

**Remark:** As noted in the introduction, the online prediction process involves using data at time $t$ to estimate the optimal parameters and using these parameters to make prediction at time $t+1$. The four terms in the upper bound can be interpreted as follows:

- The first term ($T\left(rb^2 + \frac{\gamma(G+\lambda)}{2}\right)$) relates to gradient estimation error, reflecting the imprecision in estimating optimal parameters at each step.

- The second term ($\frac{r}{\gamma}V$) measures the cumulative discrepancy between consecutive optimal parameters, indicating that even the perfect estimation at $t$ cannot be optimal at $t+1$.

- The third term ($\sum_{t=1}^{T}\mathbb{E}\left[(f(x_t|\theta_t^*) - g(x_t))^2\right]$) quantifies the approximation error between the parameterized model and the true data-generating model at the optimal parameters.

- The fourth term ($\sum_{t=1}^{T}\epsilon_t^2$) represents the inherent problem stochasticity.

**Proof C.1**

$$\frac{1}{2}\|\theta_{t+1} - \theta_t^*\|_2^2 \leq \frac{1}{2}\|\theta_t - \gamma\eta_t - \theta_t^*\|_2^2 \tag{11}$$

$$= \frac{1}{2}\|\theta_t - \theta_t^*\|_2^2 - \gamma\eta^T(\theta_t - \theta_t^*) + \frac{1}{2}\gamma^2\eta_t^2 \tag{12}$$

*Then*

$$\eta_t^T(\theta - \theta_t^*) \leq \frac{1}{2\gamma}||\theta_t - \theta_t^*||_2^2 - \frac{1}{2\gamma}||\theta_{t+1} - \theta_t^*||_2^2 + \frac{1}{2}\gamma\eta_t^2$$

$$= \frac{1}{2\gamma}||\theta_t - \theta_t^*||_2^2 - \frac{1}{2\gamma}||\theta_{t+1} - \theta_{t+1}^* + \theta_{t+1}^* - \theta_t^*||_2^2 + \frac{1}{2}\gamma\eta_t^2$$

$$= \frac{1}{2\gamma}||\theta_t - \theta_t^*||_2^2 - \frac{1}{2\gamma}||\theta_{t+1} - \theta_{t+1}^*||_2^2 - \frac{1}{2\gamma}||\theta_{t+1}^* - \theta_t^*||_2^2$$

$$+ \frac{1}{\gamma}(\theta_{t+1}^* - \theta_{t+1})^T(\theta_t^* - \theta_{t+1}^*) + \frac{1}{2}\gamma\eta_t^2$$

$$\leq \frac{1}{2\gamma}||\theta_t - \theta_t^*||_2^2 - \frac{1}{2\gamma}||\theta_{t+1} - \theta_{t+1}^*||_2^2 - \frac{1}{2\gamma}||\theta_{t+1}^* - \theta_t^*||_2^2$$

$$+ \frac{1}{\gamma}r||\theta_t^* - \theta_{t+1}^*|| + \frac{1}{2}\gamma\eta_t^2$$

$$E(f(x_t|\theta_t^*) - f(x_t|\theta_t))^2 \leq Egrad_t^T(\theta_t^* - \theta_t)$$

$$= E(b + \eta)(\theta_t^* - \theta_t)$$

$$\leq br + \frac{1}{2\gamma}||\theta_t - \theta_t^*||_2^2 - \frac{1}{2\gamma}||\theta_{t+1} - \theta_{t+1}^*||_2^2 - \frac{1}{2\gamma}||\theta_{t+1}^* - \theta_t^*||_2^2$$

$$+ \frac{1}{\gamma}r||\theta_t^* - \theta_{t+1}^*|| + E\frac{1}{2}\gamma\eta_t^2$$

*And:*

$$E\eta^2 \leq \lambda + G \tag{13}$$

*by summing the above inequalities over $t = 1, ..., T$ we have:*

$$R_d \leq Tbr + \frac{Vr}{\gamma} + \frac{T}{2}\gamma(G + \lambda)$$

Therefore, online prediction fundamentally centers on two core challenges: identifying well-suited parameters and designing effective update mechanisms. Identifying suitable parameters corresponds to minimizing the path variation $V$ (the second term), promoting stability in the optimal parameter sequence. Effective update mechanisms focus on obtaining accurate gradient estimates, minimizing $b$ and $\lambda$ (part of the first term). Techniques like experience replay or mini-batch gradient averaging offer a critical trade-off: while older data may introduce slight bias, the larger sample size significantly reduces gradient estimate variance ($\lambda$). Consequently, from an expected loss perspective, multi-sample updates might outperform single-sample updates.

We now analyze how adjustments at different levels impact performance. Comparing two approaches: full-parameter updates versus feature-only modifications. The full-parameter approach requires updating extremely high-dimensional parameters. Since the optimal parameter $V$ across time intervals corresponds to an $l_2$-norm of high-dimensional variables, larger dimensionality inherently increases $V$. Similarly, the trace of gradient covariance matrices grows proportionally with dimensionality. Feature-level adjustments require significantly fewer parameter updates, potentially reducing both $V$ and covariance trace values.

Besides, when features represent actual physical quantities (like temperatures during seasonal transitions), the $V$ might be even smaller. For example, if a feature corresponds directly to temperature and the training set consist of the data in summer and the test data is from winter, we only need consistent adjustments to that feature (just subtract some values). This consistent adjustments could keep the optimal parameter variations $V$ small.

We must emphasize that our problem is fundamentally non-convex. Furthermore, analyses comparing full parameter updates and feature-space adjustments remain conceptual rather than rigorously theoretical. Given these limitations, we include this discussion in Appendix for contextual understanding only.

# D    DISCUSSION THE BATCH SIZE USED IN COMPUTING HISTORY GRADIENT

When computing historical gradients, we employ batch data from past sequences rather than single data to mitigate high variance in gradient estimation. While batch processing reduces variance, it may introduce bias by incorporating outdated information. This necessitates balancing bias-variance trade-offs through appropriate batch size selection. In our main experiments, we adopted a batch size of 24. To validate this choice, we conducted supplementary experiments examining how varying batch sizes impact forecasting MSE during deployment.

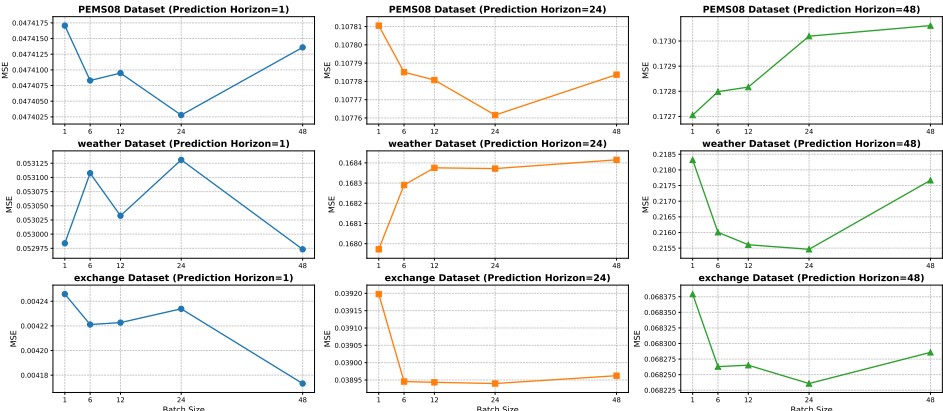

Figure 5: The relationship between batch size and final MSE across three datasets under varying prediction horizons. Each row of subplots corresponds to one dataset at different prediction horizons. And each column represents results for a fixed prediction horizon across all three datasets.

We reports iTransformer's performance across three datasets under different batch sizes at different prediction horizons (H) in Figure 5. Surprisingly, we observed no consistent optimal batch size across datasets or even within the same dataset at different horizons. As illustrated in Figure 5: For PEMS08 (top row), batch size 24 performed best at H=1 and H=24, with slight error increases at smaller/larger sizes. Conversely, at H=48, size 1 proved optimal. Weather dataset (bottom row) exhibited different patterns, demonstrating that batch size optimization is highly context-dependent – requiring case-specific tuning rather than universal rules. Finally, the differences caused by batch size are minimal, roughly at the level of the fourth decimal place.

# E    SENSITIVE ANALYSIS OF ONLINE LEARNING RATE

The learning rate for online updates is another important hyperparameter in our experiments, so we conducted a sensitivity analysis on it. In our main experiments, we set this parameter to 0.0003. To further investigate, we performed additional tests using values of 0.00005, 0.0001, 0.0003, 0.0005, 0.001, and 0.005. Note that we only adjusted the online learning rate for the adapter module, while the online learning rate for the final layer was consistently kept at 0.00003.

The results are shown in Figure 6. The red dots highlight the learning rates that achieved the minimum MSE. We observed that the best performance mostly occurred within the range of 0.00005 to 0.0003, with only minor differences between them. However, it is important to note that when the online learning rate exceeded 0.005, the MSE increased significantly.

# F    FULL RESULTS

## F.1    FULL RESULTS OF MAIN EXPERIMENTS

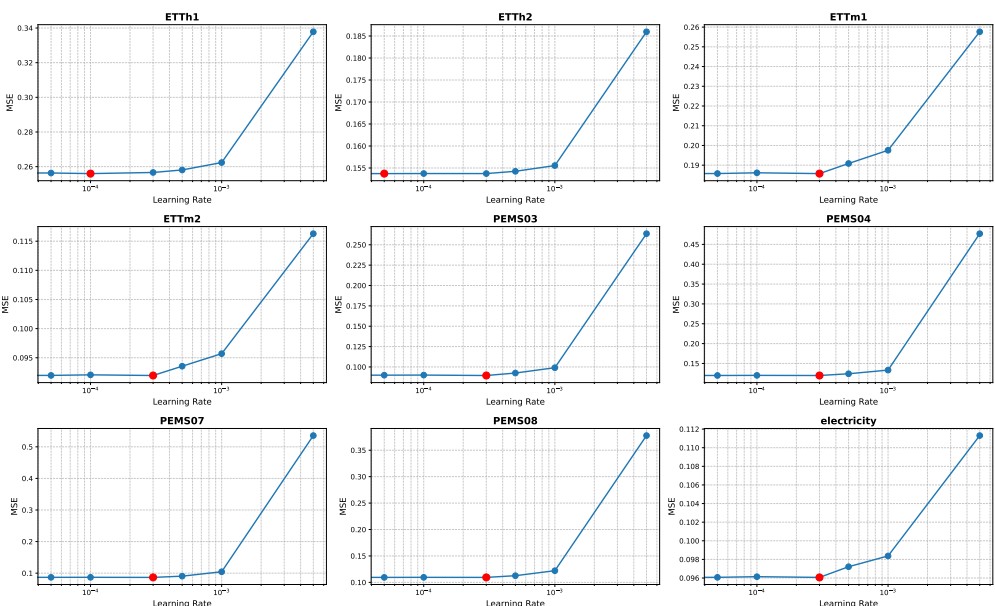

Figure 6: Sensitive analysis of online learning rate

Table 8: The full results of online prediction experiments

| Base Model | | iTransformer | | | | | | | | | SOFTS | | | | | | | | | TimesNet | | | | | | | | | | |
|---|---|---|---|---|---|---|---|---|---|---|---|---|---|---|---|---|---|---|---|---|---|---|---|---|---|---|---|---|---|---|
| Dataset | H | Ori | fOGD | OGD | DSOF | SOLID | ADCSD | Proceed | ADAPT-Z | IMP | Ori | fOGD | OGD | DSOF | SOLID | ADCSD | Proceed | ADAPT-Z | IMP | Ori | fOGD | OGD | DSOF | SOLID | ADCSD | Proceed | ADAPT-Z | IMP | FSNET | Onenet |
| ETTh1 | 1 | 0.1228 | 0.1196 | 0.1217 | 0.1229 | 0.1217 | 0.1210 | 0.1224 | 0.1127 | 8.21% | 0.1211 | 0.1201 | 0.1190 | 0.1315 | 0.1196 | 0.1206 | 0.1207 | 0.1119 | 7.60% | 0.1383 | 0.1381 | 0.1364 | 0.1610 | 0.1380 | 0.1377 | 0.1381 | 0.1266 | 8.44% | 0.9359 | 0.7790 |
| | 24 | 0.3325 | 0.3293 | 0.3320 | 0.3360 | 0.3222 | 0.3321 | 0.3223 | 0.3126 | 5.99% | 0.3201 | 0.3199 | 0.3216 | 0.3340 | 0.3201 | 0.3201 | 0.3202 | 0.3031 | 5.29% | 0.3437 | 0.3438 | 0.3429 | 0.4542 | 0.3438 | 0.3435 | 0.3438 | 0.3304 | 3.87% | 1.0510 | 0.7780 |
| | 48 | 0.3714 | 0.3702 | 0.3712 | 0.3911 | 0.3680 | 0.3710 | 0.3680 | 0.3626 | 2.36% | 0.3632 | 0.3624 | 0.3648 | 0.3914 | 0.3633 | 0.3632 | 0.3633 | 0.3548 | 2.30% | 0.3904 | 0.3905 | 0.3893 | 0.5760 | 0.3906 | 0.3900 | 0.3905 | 0.3767 | 3.52% | 1.0792 | 0.7850 |
| ETTh2 | 1 | 0.0696 | 0.0691 | 0.0695 | 0.0684 | 0.0684 | 0.0693 | 0.0683 | 0.0674 | 3.15% | 0.0699 | 0.0698 | 0.0692 | 0.0695 | 0.0698 | 0.0699 | 0.0699 | 0.0659 | 5.69% | 0.0764 | 0.0773 | 0.0773 | 0.0796 | 0.0776 | 0.0765 | 0.0773 | 0.0745 | 2.52% | 1.0449 | 0.9573 |
| | 24 | 0.1827 | 0.1824 | 0.1827 | 0.1873 | 0.1830 | 0.1825 | 0.1828 | 0.1771 | 3.07% | 0.1796 | 0.1792 | 0.1795 | 0.1841 | 0.1797 | 0.1796 | 0.1797 | 0.1714 | 4.59% | 0.1982 | 0.1986 | 0.1985 | 0.2672 | 0.1984 | 0.1982 | 0.1986 | 0.1947 | 1.78% | 1.1279 | 1.0204 |
| | 48 | 0.2382 | 0.2382 | 0.2382 | 0.2382 | 0.2353 | 0.2381 | 0.2353 | 0.2302 | 3.35% | 0.2306 | 0.2304 | 0.2306 | 0.2894 | 0.2306 | 0.2306 | 0.2306 | 0.2239 | 2.92% | 0.2383 | 0.2382 | 0.2382 | 0.3500 | 0.2382 | 0.2382 | 0.2382 | 0.2381 | 0.04% | 1.1601 | 1.0211 |
| ETTm1 | 1 | 0.0570 | 0.0566 | 0.0568 | 0.0793 | 0.0565 | 0.0567 | 0.0566 | 0.0545 | 4.42% | 0.0564 | 0.0551 | 0.0551 | 0.0734 | 0.0558 | 0.0554 | 0.0564 | 0.0528 | 6.41% | 0.0583 | 0.0583 | 0.0579 | 0.0767 | 0.0582 | 0.0583 | 0.0583 | 0.0555 | 4.83% | 0.9022 | 0.9863 |
| | 24 | 0.2702 | 0.2624 | 0.2680 | 0.2928 | 0.2547 | 0.2659 | 0.2548 | 0.2293 | 15.15% | 0.2405 | 0.2379 | 0.2389 | 0.2885 | 0.2403 | 0.2391 | 0.2405 | 0.2166 | 9.91% | 0.2713 | 0.2713 | 0.2605 | 0.3496 | 0.2710 | 0.2651 | 0.2713 | 0.2420 | 10.81% | 0.9943 | 1.0165 |
| | 48 | 0.3655 | 0.3508 | 0.3624 | 0.3838 | 0.3421 | 0.3518 | 0.3421 | 0.3024 | 17.27% | 0.3126 | 0.3096 | 0.3129 | 0.3971 | 0.3125 | 0.3104 | 0.3126 | 0.2877 | 7.95% | 0.3584 | 0.3584 | 0.3492 | 0.4408 | 0.3582 | 0.3494 | 0.3584 | 0.3022 | 15.67% | 1.0452 | 1.0209 |
| ETTm2 | 1 | 0.0327 | 0.0325 | 0.0326 | 0.0480 | 0.0326 | 0.0326 | 0.0326 | 0.0318 | 2.73% | 0.0335 | 0.0333 | 0.0334 | 0.0452 | 0.0336 | 0.0334 | 0.0335 | 0.0319 | 4.80% | 0.0332 | 0.0332 | 0.0331 | 0.0445 | 0.0332 | 0.0332 | 0.0332 | 0.0324 | 2.55% | 1.0206 | 0.9076 |
| | 24 | 0.1134 | 0.1129 | 0.1134 | 0.1456 | 0.1144 | 0.1130 | 0.1144 | 0.1082 | 4.62% | 0.1073 | 0.1071 | 0.1075 | 0.1539 | 0.1073 | 0.1070 | 0.1073 | 0.1037 | 3.37% | 0.1139 | 0.1139 | 0.1137 | 0.1650 | 0.1140 | 0.1137 | 0.1139 | 0.1073 | 5.76% | 1.0597 | 0.9633 |
| | 48 | 0.1449 | 0.1447 | 0.1449 | 0.2104 | 0.1450 | 0.1446 | 0.1450 | 0.1423 | 1.79% | 0.1430 | 0.1426 | 0.1431 | 0.2084 | 0.1429 | 0.1427 | 0.1430 | 0.1403 | 1.86% | 0.1536 | 0.1536 | 0.1533 | 0.2439 | 0.1536 | 0.1532 | 0.1536 | 0.1455 | 5.28% | 1.0875 | 0.9921 |
| PEMS03 | 1 | 0.0388 | 0.0385 | 0.0387 | 0.0441 | 0.0387 | 0.0386 | 0.0388 | 0.0381 | 1.67% | 0.0390 | 0.0387 | 0.0390 | 0.0454 | 0.0389 | 0.0389 | 0.0390 | 0.0382 | 1.97% | 0.0593 | 0.0594 | 0.0594 | 0.0505 | 0.0594 | 0.0592 | 0.0594 | 0.0582 | 1.94% | 0.3955 | 0.3947 |
| | 24 | 0.0974 | 0.0938 | 0.0971 | 0.1829 | 0.1017 | 0.0968 | 0.1017 | 0.0922 | 5.26% | 0.0912 | 0.0897 | 0.0923 | 0.1795 | 0.0915 | 0.0908 | 0.0912 | 0.0892 | 2.19% | 0.1023 | 0.1023 | 0.1021 | 0.1282 | 0.1025 | 0.1019 | 0.1023 | 0.0986 | 3.62% | 0.4065 | 0.4049 |
| | 48 | 0.1691 | 0.1637 | 0.1689 | 0.3113 | 0.1776 | 0.1668 | 0.1777 | 0.1617 | 4.38% | 0.1419 | 0.1412 | 0.1443 | 0.3301 | 0.1419 | 0.1415 | 0.1420 | 0.1411 | 0.58% | 0.1496 | 0.1504 | 0.1500 | 0.2080 | 0.1505 | 0.1493 | 0.1504 | 0.1458 | 2.55% | 0.4299 | 0.4282 |
| PEMS04 | 1 | 0.0566 | 0.0561 | 0.0562 | 0.0724 | 0.0564 | 0.0566 | 0.0567 | 0.0554 | 2.24% | 0.0557 | 0.0555 | 0.0558 | 0.0619 | 0.0557 | 0.0558 | 0.0558 | 0.0550 | 1.20% | 0.0795 | 0.0797 | 0.0797 | 0.0635 | 0.0797 | 0.0796 | 0.0797 | 0.0788 | 0.88% | 0.4631 | 0.4633 |
| | 24 | 0.1319 | 0.1268 | 0.1309 | 0.1596 | 0.1315 | 0.1279 | 0.1316 | 0.1214 | 8.01% | 0.1188 | 0.1196 | 0.1203 | 0.1527 | 0.1192 | 0.1186 | 0.1190 | 0.1176 | 0.98% | 0.1246 | 0.1247 | 0.1247 | 0.1302 | 0.1249 | 0.1246 | 0.1247 | 0.1228 | 1.44% | 0.4740 | 0.4735 |
| | 48 | 0.2305 | 0.2201 | 0.2283 | 0.2586 | 0.2317 | 0.2280 | 0.2318 | 0.2024 | 12.19% | 0.1965 | 0.1884 | 0.1964 | 0.2515 | 0.1965 | 0.1961 | 0.1965 | 0.1862 | 5.23% | 0.1654 | 0.1654 | 0.1646 | 0.1676 | 0.1659 | 0.1650 | 0.1654 | 0.1614 | 2.42% | 0.4863 | 0.4820 |
| PEMS07 | 1 | 0.0408 | 0.0389 | 0.0396 | 0.0444 | 0.0404 | 0.0399 | 0.0407 | 0.0389 | 4.50% | 0.0365 | 0.0362 | 0.0365 | 0.0390 | 0.0365 | 0.0364 | 0.0365 | 0.0360 | 1.40% | 0.0639 | 0.0638 | 0.0638 | 0.0462 | 0.0638 | 0.0634 | 0.0638 | 0.0611 | 4.32% | 0.4816 | 0.4814 |
| | 24 | 0.0938 | 0.0904 | 0.0936 | 0.1221 | 0.0938 | 0.0922 | 0.0939 | 0.0899 | 4.15% | 0.0893 | 0.0886 | 0.0900 | 0.1171 | 0.0893 | 0.0887 | 0.0893 | 0.0880 | 1.45% | 0.0960 | 0.0960 | 0.0960 | 0.0974 | 0.0960 | 0.0957 | 0.0960 | 0.0928 | 3.36% | 0.4843 | 0.4844 |
| | 48 | 0.1450 | 0.1393 | 0.1447 | 0.2052 | 0.1447 | 0.1407 | 0.1447 | 0.1386 | 4.36% | 0.1375 | 0.1389 | 0.1393 | 0.1952 | 0.1376 | 0.1370 | 0.1376 | 0.1356 | 1.43% | 0.1254 | 0.1252 | 0.1252 | 0.1346 | 0.1252 | 0.1244 | 0.1255 | 0.1220 | 2.67% | 0.4920 | 0.4894 |
| PEMS08 | 1 | 0.0527 | 0.0513 | 0.0519 | 0.0661 | 0.0521 | 0.0523 | 0.0528 | 0.0502 | 4.75% | 0.0489 | 0.0483 | 0.0487 | 0.0560 | 0.0488 | 0.0488 | 0.0490 | 0.0474 | 3.05% | 0.1053 | 0.1054 | 0.1054 | 0.0622 | 0.1054 | 0.1053 | 0.1054 | 0.1027 | 2.44% | 0.5953 | 0.5954 |
| | 24 | 0.1285 | 0.1253 | 0.1282 | 0.1517 | 0.1324 | 0.1271 | 0.1325 | 0.1176 | 8.49% | 0.1126 | 0.1097 | 0.1138 | 0.1480 | 0.1126 | 0.1124 | 0.1126 | 0.1078 | 4.27% | 0.1788 | 0.1790 | 0.1782 | 0.2039 | 0.1789 | 0.1787 | 0.1790 | 0.1718 | 3.91% | 0.6068 | 0.6080 |
| | 48 | 0.2922 | 0.2871 | 0.2914 | 0.3179 | 0.2961 | 0.2867 | 0.2962 | 0.2682 | 8.20% | 0.1771 | 0.1753 | 0.1792 | 0.2586 | 0.1771 | 0.1768 | 0.1771 | 0.1730 | 2.32% | 0.2523 | 0.2523 | 0.2505 | 0.3464 | 0.2523 | 0.2516 | 0.2523 | 0.2449 | 2.93% | 0.6236 | 0.6220 |
| electricity | 1 | 0.0534 | 0.0510 | 0.0512 | 0.0614 | 0.0526 | 0.0518 | 0.0533 | 0.0484 | 9.20% | 0.0485 | 0.0478 | 0.0485 | 0.0524 | 0.0483 | 0.0482 | 0.0484 | 0.0473 | 2.41% | 0.1214 | 0.1211 | 0.1210 | 0.0639 | 0.1211 | 0.1206 | 0.1211 | 0.1142 | 5.98% | 0.7763 | 0.7790 |
| | 24 | 0.1126 | 0.1109 | 0.1124 | 0.1275 | 0.1126 | 0.1119 | 0.1126 | 0.1089 | 3.24% | 0.1104 | 0.1098 | 0.1117 | 0.1199 | 0.1104 | 0.1102 | 0.1104 | 0.1090 | 1.27% | 0.1438 | 0.1438 | 0.1437 | 0.1379 | 0.1437 | 0.1435 | 0.1438 | 0.1382 | 3.95% | 0.7790 | 0.7780 |
| | 48 | 0.1386 | 0.1358 | 0.1385 | 0.1480 | 0.1385 | 0.1381 | 0.1385 | 0.1339 | 3.39% | 0.1338 | 0.1317 | 0.1348 | 0.1426 | 0.1334 | 0.1330 | 0.1333 | 0.1318 | 1.46% | 0.1619 | 0.1613 | 0.1609 | 0.1531 | 0.1617 | 0.1610 | 0.1613 | 0.1550 | 4.29% | 0.7854 | 0.7850 |
| exchange | 1 | 0.0047 | 0.0046 | 0.0046 | 0.0046 | 0.0044 | 0.0046 | 0.0044 | 0.0042 | 9.78% | 0.0044 | 0.0044 | 0.0044 | 0.0124 | 0.0044 | 0.0044 | 0.0043 | 0.0042 | 3.14% | 0.0102 | 0.0103 | 0.0103 | 0.0074 | 0.0103 | 0.0103 | 0.0103 | 0.0104 | -1.50% | 2.3293 | 1.2223 |
| | 24 | 0.0426 | 0.0420 | 0.0426 | 0.0456 | 0.0416 | 0.0423 | 0.0417 | 0.0404 | 5.03% | 0.0401 | 0.0402 | 0.0403 | 0.0508 | 0.0402 | 0.0402 | 0.0402 | 0.0389 | 2.87% | 0.0564 | 0.0558 | 0.0558 | 0.0519 | 0.0558 | 0.0558 | 0.0558 | 0.0559 | 0.79% | 2.3467 | 1.2281 |
| | 48 | 0.0709 | 0.0710 | 0.0709 | 0.0770 | 0.0715 | 0.0709 | 0.0719 | 0.0706 | 0.45% | 0.0683 | 0.0684 | 0.0681 | 0.0929 | 0.0683 | 0.0681 | 0.0681 | 0.0682 | 0.07% | 0.0926 | 0.0922 | 0.0922 | 0.1015 | 0.0923 | 0.0921 | 0.0922 | 0.0933 | -0.79% | 2.4027 | 1.2411 |
| solar | 1 | 0.0087 | 0.0084 | 0.0084 | 0.0089 | 0.0086 | 0.0086 | 0.0088 | 0.0077 | 10.75% | 0.0076 | 0.0072 | 0.0072 | 0.0079 | 0.0075 | 0.0075 | 0.0076 | 0.0069 | 8.56% | 0.0289 | 0.0292 | 0.0290 | 0.0087 | 0.0292 | 0.0290 | 0.0290 | 0.0284 | 1.59% | 0.0637 | 0.1173 |
| | 24 | 0.1231 | 0.1190 | 0.1201 | 0.1109 | 0.1264 | 0.1191 | 0.1266 | 0.0972 | 21.05% | 0.0969 | 0.0970 | 0.0970 | 0.1075 | 0.0970 | 0.0970 | 0.0971 | 0.0881 | 9.04% | 0.1336 | 0.1354 | 0.1311 | 0.1187 | 0.1356 | 0.1317 | 0.1354 | 0.1249 | 6.53% | 0.1189 | 0.2133 |
| | 48 | 0.2393 | 0.2293 | 0.2346 | 0.1828 | 0.2300 | 0.2350 | 0.2302 | 0.1771 | 25.99% | 0.1542 | 0.1556 | 0.1553 | 0.1830 | 0.1548 | 0.1543 | 0.1544 | 0.1423 | 7.71% | 0.1837 | 0.1854 | 0.1820 | 0.2058 | 0.1854 | 0.1855 | 0.1854 | 0.1802 | 1.93% | 0.1545 | 0.2592 |
| traffic | 1 | 0.2132 | 0.2097 | 0.2108 | 0.2992 | 0.2103 | 0.2121 | 0.2133 | 0.2003 | 6.05% | 0.2009 | 0.2001 | 0.2010 | 0.2346 | 0.2007 | 0.2010 | 0.2013 | 0.1959 | 2.52% | 0.5060 | 0.5065 | 0.4912 | 0.2872 | 0.5035 | 0.5048 | 0.5065 | 0.3960 | 21.73% | 0.8288 | 0.4926 |
| | 24 | 0.4188 | 0.4143 | 0.4169 | 0.4173 | 0.4179 | 0.4181 | 0.4185 | 0.3820 | 8.80% | 0.3559 | 0.3568 | 0.3584 | 0.4130 | 0.3560 | 0.3559 | 0.3561 | 0.3520 | 1.08% | 0.5644 | 0.5647 | 0.5525 | 0.5262 | 0.5640 | 0.5629 | 0.5647 | 0.4939 | 12.49% | 0.8791 | 0.4917 |
| | 48 | 0.4328 | 0.4298 | 0.4324 | 0.4462 | 0.4329 | 0.4316 | 0.4331 | 0.4120 | 4.82% | 0.3879 | 0.3898 | 0.3916 | 0.4421 | 0.3884 | 0.3883 | 0.3885 | 0.3848 | 0.80% | 0.5878 | 0.5895 | 0.5753 | 0.5878 | 0.5889 | 0.5885 | 0.5896 | 0.5027 | 14.46% | 0.9059 | 0.5168 |
| weather | 1 | 0.0547 | 0.0542 | 0.0540 | 0.0596 | 0.0565 | 0.0577 | 0.0579 | 0.0505 | 7.67% | 0.0591 | 0.0589 | 0.0588 | 0.0596 | 0.0591 | 0.0591 | 0.0593 | 0.0531 | 10.17% | 0.0659 | 0.0660 | 0.0659 | 0.0697 | 0.0659 | 0.0658 | 0.0660 | 0.0645 | 2.08% | 0.6021 | 0.5375 |
| | 24 | 0.1878 | 0.1847 | 0.1868 | 0.1816 | 0.3007 | 0.1774 | 0.1819 | 0.1730 | 7.92% | 0.1695 | 0.1692 | 0.1706 | 0.2098 | 0.1697 | 0.1695 | 0.1697 | 0.1675 | 1.18% | 0.1840 | 0.1842 | 0.1838 | 0.2819 | 0.1842 | 0.1838 | 0.1842 | 0.1765 | 4.05% | 0.6136 | 0.5512 |
| | 48 | 0.2225 | 0.2204 | 0.2224 | 0.2632 | 0.2239 | 0.2211 | 0.2240 | 0.2147 | 3.49% | 0.2426 | 0.2472 | 0.2470 | 0.2628 | 0.2433 | 0.2426 | 0.2428 | 0.2155 | 11.19% | 0.2312 | 0.2314 | 0.2309 | 0.3625 | 0.2313 | 0.2306 | 0.2314 | 0.2164 | 6.39% | 0.6191 | 0.5583 |

According to Table 8, our method consistently delivers the best results when applied to iTransformer and SOFTS models under all scenarios. However, we see different patterns with TimesNet. While our approach usually ranks first with TimesNet too, it achieves second-best results in some cases.

We suspect TimesNet itself has limitations. Without online adaptation, TimesNet often underperforms compared to iTransformer and SOFTS, especially in one-step predictions. For example, on PEMS03 dataset with one-step prediction, TimesNet's MSE reached 0.0593, while the other two models achieved 0.0385 and 0.0387. Similarly for electricity dataset with one-step prediction, TimesNet scored 0.1214 MSE versus 0.0534 and 0.0477. Our method performs poorly exactly under these conditions where TimesNet's original predictions are weak. This suggests that the features TimesNet produces in such cases may lack sufficient quality for effective adaptation.

Additionally, DSOF significantly outperforms other methods in these challenging TimesNet scenarios. The reason may be that DSOF's adapter directly accesses raw input data, allowing it to link inputs and true values. In contrast, our method must rely solely on features extracted by the model, which might lose valuable information. Similarly, ADCSD adjusts outputs using the model's results, but if the outputs already miss key details, improvement becomes difficult. As for SOLID, this method updates parameter in the final layer of TimesNet, it might simply be insufficient to overcome TimesNet's fundamental limitations.

### F.2    RESULTS OF EXPERIMENTS USING DIFFERENT SEEDS

Our experimental results with the iTransformer model under different random seeds are presented in Table 9. Across all tested random seeds (2024, 2025, 2026), our approach maintains superior forecasting accuracy compared to baseline methods.

Table 9: The results of online prediction experiments with different seeds

| Dataset | H | 2024 Ori | fOGD | OGD | DSOF | SOLID | ADCSD | Proceed | ADAPT-Z | IMP | 2025 Ori | fOGD | OGD | DSOF | SOLID | ADCSD | Proceed | ADAPT-Z | IMP | 2026 Ori | fOGD | OGD | DSOF | SOLID | ADCSD | Proceed | ADAPT-Z | IMP |
|---|---|---|---|---|---|---|---|---|---|---|---|---|---|---|---|---|---|---|---|---|---|---|---|---|---|---|---|---|
| ETTh1 | 1 | 0.1270 | 0.1260 | 0.1263 | 0.1229 | 0.1218 | 0.1267 | 0.1223 | 0.1186 | 6.59% | 0.1228 | 0.1196 | 0.1217 | 0.1229 | 0.1217 | 0.1210 | 0.1224 | 0.1127 | 8.21% | 0.1272 | 0.1244 | 0.1251 | 0.1220 | 0.1269 | 0.1261 | 0.1278 | 0.1175 | 7.59% |
| | 24 | 0.3340 | 0.3328 | 0.3336 | 0.3337 | 0.3294 | 0.3340 | 0.3295 | 0.3156 | 5.52% | 0.3325 | 0.3293 | 0.3320 | 0.3360 | 0.3222 | 0.3321 | 0.3223 | 0.3126 | 5.99% | 0.3313 | 0.3279 | 0.3309 | 0.3343 | 0.3238 | 0.3312 | 0.3239 | 0.3139 | 5.25% |
| | 48 | 0.3723 | 0.3699 | 0.3722 | 0.3871 | 0.3666 | 0.3724 | 0.3666 | 0.3597 | 3.39% | 0.3714 | 0.3702 | 0.3712 | 0.3911 | 0.3680 | 0.3710 | 0.3680 | 0.3626 | 2.36% | 0.3854 | 0.3820 | 0.3852 | 0.3867 | 0.3788 | 0.3853 | 0.3787 | 0.3670 | 4.77% |
| ETTh2 | 1 | 0.0753 | 0.0748 | 0.0750 | 0.0715 | 0.0733 | 0.0752 | 0.0734 | 0.0715 | 4.97% | 0.0696 | 0.0691 | 0.0695 | 0.0684 | 0.0684 | 0.0693 | 0.0683 | 0.0674 | 3.15% | 0.0707 | 0.0706 | 0.0707 | 0.0672 | 0.0705 | 0.0708 | 0.0706 | 0.0700 | 1.06% |
| | 24 | 0.1817 | 0.1815 | 0.1818 | 0.1856 | 0.1780 | 0.1818 | 0.1778 | 0.1741 | 4.19% | 0.1827 | 0.1824 | 0.1827 | 0.1873 | 0.1830 | 0.1825 | 0.1828 | 0.1771 | 3.07% | 0.1837 | 0.1837 | 0.1838 | 0.1873 | 0.1804 | 0.1838 | 0.1804 | 0.1778 | 3.24% |
| | 48 | 0.2304 | 0.2304 | 0.2304 | 0.2576 | 0.2292 | 0.2304 | 0.2292 | 0.2263 | 1.80% | 0.2382 | 0.2382 | 0.2382 | 0.2382 | 0.2353 | 0.2381 | 0.2353 | 0.2302 | 3.35% | 0.2339 | 0.2335 | 0.2339 | 0.2433 | 0.2273 | 0.2339 | 0.2273 | 0.2280 | 2.51% |
| ETTm1 | 1 | 0.0578 | 0.0569 | 0.0568 | 0.0783 | 0.0566 | 0.0575 | 0.0569 | 0.0540 | 6.52% | 0.0570 | 0.0566 | 0.0568 | 0.0793 | 0.0565 | 0.0567 | 0.0566 | 0.0545 | 4.42% | 0.0590 | 0.0586 | 0.0576 | 0.0780 | 0.0575 | 0.0589 | 0.0578 | 0.0544 | 7.77% |
| | 24 | 0.2707 | 0.2612 | 0.2678 | 0.3050 | 0.2618 | 0.2661 | 0.2619 | 0.2306 | 14.79% | 0.2702 | 0.2624 | 0.2680 | 0.2928 | 0.2547 | 0.2659 | 0.2548 | 0.2293 | 15.15% | 0.2622 | 0.2516 | 0.2587 | 0.2963 | 0.2548 | 0.2576 | 0.2549 | 0.2275 | 13.23% |
| | 48 | 0.3352 | 0.3290 | 0.3339 | 0.3936 | 0.3209 | 0.3319 | 0.3207 | 0.2963 | 11.59% | 0.3655 | 0.3508 | 0.3624 | 0.3838 | 0.3421 | 0.3518 | 0.3421 | 0.3024 | 17.27% | 0.3351 | 0.3280 | 0.3335 | 0.3987 | 0.3172 | 0.3323 | 0.3173 | 0.2961 | 11.63% |
| ETTm2 | 1 | 0.0338 | 0.0335 | 0.0337 | 0.0481 | 0.0338 | 0.0336 | 0.0336 | 0.0324 | 4.13% | 0.0327 | 0.0325 | 0.0326 | 0.0480 | 0.0326 | 0.0326 | 0.0326 | 0.0318 | 2.73% | 0.0339 | 0.0335 | 0.0336 | 0.0480 | 0.0337 | 0.0337 | 0.0337 | 0.0321 | 5.35% |
| | 24 | 0.1106 | 0.1103 | 0.1106 | 0.1487 | 0.1109 | 0.1106 | 0.1109 | 0.1066 | 5.30% | 0.1134 | 0.1129 | 0.1134 | 0.1456 | 0.1144 | 0.1130 | 0.1144 | 0.1082 | 4.62% | 0.1132 | 0.1127 | 0.1131 | 0.1491 | 0.1173 | 0.1128 | 0.1171 | 0.1081 | 4.54% |
| | 48 | 0.1453 | 0.1451 | 0.1452 | 0.1890 | 0.1453 | 0.1453 | 0.1452 | 0.1426 | 1.82% | 0.1449 | 0.1447 | 0.1449 | 0.2104 | 0.1450 | 0.1446 | 0.1450 | 0.1423 | 1.79% | 0.1486 | 0.1481 | 0.1485 | 0.2037 | 0.1490 | 0.1483 | 0.1489 | 0.1441 | 3.05% |
| PEMS03 | 1 | 0.0411 | 0.0396 | 0.0393 | 0.0441 | 0.0402 | 0.0406 | 0.0412 | 0.0381 | 7.33% | 0.0388 | 0.0385 | 0.0387 | 0.0441 | 0.0387 | 0.0386 | 0.0388 | 0.0381 | 1.67% | 0.0392 | 0.0388 | 0.0391 | 0.0438 | 0.0391 | 0.0391 | 0.0392 | 0.0380 | 2.97% |
| | 24 | 0.0924 | 0.0895 | 0.0923 | 0.1851 | 0.0935 | 0.0920 | 0.0935 | 0.0895 | 3.12% | 0.0974 | 0.0938 | 0.0971 | 0.1829 | 0.1017 | 0.0968 | 0.1017 | 0.0922 | 5.26% | 0.0965 | 0.0936 | 0.0963 | 0.1953 | 0.1002 | 0.0959 | 0.1002 | 0.0926 | 4.01% |
| | 48 | 0.1772 | 0.1719 | 0.1768 | 0.3275 | 0.1795 | 0.1765 | 0.1793 | 0.1700 | 4.02% | 0.1691 | 0.1637 | 0.1689 | 0.3113 | 0.1776 | 0.1668 | 0.1777 | 0.1617 | 4.38% | 0.2129 | 0.2067 | 0.2107 | 0.3378 | 0.2154 | 0.2116 | 0.2155 | 0.1948 | 8.49% |
| PEMS04 | 1 | 0.0580 | 0.0571 | 0.0569 | 0.0711 | 0.0573 | 0.0581 | 0.0580 | 0.0556 | 4.07% | 0.0566 | 0.0561 | 0.0562 | 0.0724 | 0.0564 | 0.0566 | 0.0567 | 0.0554 | 2.24% | 0.0608 | 0.0595 | 0.0603 | 0.0715 | 0.0605 | 0.0609 | 0.0610 | 0.0594 | 2.39% |
| | 24 | 0.1226 | 0.1191 | 0.1226 | 0.1587 | 0.1231 | 0.1225 | 0.1231 | 0.1187 | 3.18% | 0.1319 | 0.1268 | 0.1309 | 0.1596 | 0.1315 | 0.1279 | 0.1316 | 0.1214 | 8.01% | 0.1232 | 0.1199 | 0.1233 | 0.1583 | 0.1236 | 0.1232 | 0.1236 | 0.1196 | 2.94% |
| | 48 | 0.1900 | 0.1839 | 0.1900 | 0.2623 | 0.1912 | 0.1894 | 0.1911 | 0.1861 | 2.03% | 0.2305 | 0.2201 | 0.2283 | 0.2586 | 0.2317 | 0.2280 | 0.2318 | 0.2024 | 12.19% | 0.2611 | 0.2545 | 0.2608 | 0.2693 | 0.2611 | 0.2604 | 0.2612 | 0.2539 | 2.75% |
| PEMS07 | 1 | 0.0382 | 0.0372 | 0.0377 | 0.0450 | 0.0378 | 0.0379 | 0.0381 | 0.0370 | 3.04% | 0.0408 | 0.0389 | 0.0396 | 0.0444 | 0.0404 | 0.0399 | 0.0407 | 0.0389 | 4.50% | 0.0394 | 0.0371 | 0.0374 | 0.0439 | 0.0385 | 0.0387 | 0.0396 | 0.0358 | 9.09% |
| | 24 | 0.1049 | 0.0991 | 0.1046 | 0.1242 | 0.1049 | 0.1042 | 0.1050 | 0.0985 | 6.18% | 0.0938 | 0.0904 | 0.0936 | 0.1221 | 0.0938 | 0.0922 | 0.0939 | 0.0899 | 4.15% | 0.0952 | 0.0923 | 0.0952 | 0.1227 | 0.0951 | 0.0950 | 0.0952 | 0.0920 | 3.42% |
| | 48 | 0.1487 | 0.1397 | 0.1487 | 0.2017 | 0.1487 | 0.1476 | 0.1487 | 0.1428 | 3.92% | 0.1450 | 0.1393 | 0.1447 | 0.2052 | 0.1447 | 0.1407 | 0.1447 | 0.1386 | 4.36% | 0.1456 | 0.1415 | 0.1455 | 0.2012 | 0.1457 | 0.1449 | 0.1457 | 0.1410 | 3.18% |
| PEMS08 | 1 | 0.0502 | 0.0498 | 0.0500 | 0.0655 | 0.0503 | 0.0502 | 0.0507 | 0.0489 | 2.71% | 0.0527 | 0.0513 | 0.0519 | 0.0661 | 0.0521 | 0.0523 | 0.0528 | 0.0502 | 4.75% | 0.0516 | 0.0507 | 0.0508 | 0.0655 | 0.0510 | 0.0516 | 0.0518 | 0.0487 | 5.53% |
| | 24 | 0.1411 | 0.1354 | 0.1384 | 0.1488 | 0.1421 | 0.1398 | 0.1423 | 0.1220 | 13.52% | 0.1285 | 0.1253 | 0.1282 | 0.1517 | 0.1324 | 0.1271 | 0.1325 | 0.1176 | 8.49% | 0.1528 | 0.1489 | 0.1519 | 0.1537 | 0.1540 | 0.1525 | 0.1541 | 0.1421 | 6.95% |
| | 48 | 0.3258 | 0.3174 | 0.3225 | 0.3015 | 0.3229 | 0.3252 | 0.3231 | 0.2890 | 11.29% | 0.2922 | 0.2871 | 0.2914 | 0.3179 | 0.2961 | 0.2867 | 0.2962 | 0.2682 | 8.20% | 0.2381 | 0.2329 | 0.2377 | 0.2763 | 0.2343 | 0.2377 | 0.2343 | 0.2241 | 5.90% |
| electricity | 1 | 0.0537 | 0.0519 | 0.0527 | 0.0615 | 0.0531 | 0.0534 | 0.0536 | 0.0489 | 8.92% | 0.0534 | 0.0510 | 0.0512 | 0.0614 | 0.0526 | 0.0518 | 0.0533 | 0.0484 | 9.20% | 0.0544 | 0.0524 | 0.0530 | 0.0607 | 0.0538 | 0.0541 | 0.0543 | 0.0492 | 9.47% |
| | 24 | 0.1122 | 0.1103 | 0.1120 | 0.1277 | 0.1122 | 0.1120 | 0.1122 | 0.1084 | 3.37% | 0.1126 | 0.1109 | 0.1124 | 0.1275 | 0.1126 | 0.1119 | 0.1126 | 0.1089 | 3.24% | 0.1145 | 0.1126 | 0.1143 | 0.1281 | 0.1145 | 0.1143 | 0.1145 | 0.1102 | 3.74% |
| | 48 | 0.1345 | 0.1314 | 0.1341 | 0.1490 | 0.1342 | 0.1339 | 0.1341 | 0.1308 | 2.79% | 0.1386 | 0.1358 | 0.1385 | 0.1480 | 0.1385 | 0.1381 | 0.1385 | 0.1339 | 3.39% | 0.1333 | 0.1304 | 0.1328 | 0.1489 | 0.1328 | 0.1327 | 0.1329 | 0.1301 | 2.41% |
| exchange | 1 | 0.0044 | 0.0044 | 0.0044 | 0.0048 | 0.0044 | 0.0044 | 0.0044 | 0.0043 | 1.40% | 0.0047 | 0.0046 | 0.0046 | 0.0046 | 0.0044 | 0.0046 | 0.0044 | 0.0042 | 9.78% | 0.0047 | 0.0046 | 0.0048 | 0.0046 | 0.0046 | 0.0048 | 0.0046 | 0.0043 | 7.81% |
| | 24 | 0.0402 | 0.0404 | 0.0403 | 0.0432 | 0.0397 | 0.0403 | 0.0397 | 0.0399 | 0.72% | 0.0426 | 0.0420 | 0.0426 | 0.0456 | 0.0416 | 0.0423 | 0.0417 | 0.0404 | 5.03% | 0.0433 | 0.0437 | 0.0436 | 0.0430 | 0.0450 | 0.0436 | 0.0450 | 0.0420 | 3.05% |
| | 48 | 0.0718 | 0.0719 | 0.0716 | 0.0801 | 0.0724 | 0.0716 | 0.0722 | 0.0706 | 1.72% | 0.0709 | 0.0710 | 0.0709 | 0.0770 | 0.0715 | 0.0709 | 0.0719 | 0.0706 | 0.45% | 0.0720 | 0.0720 | 0.0718 | 0.0859 | 0.0723 | 0.0718 | 0.0723 | 0.0713 | 1.01% |
| solar | 1 | 0.0091 | 0.0088 | 0.0087 | 0.0089 | 0.0095 | 0.0089 | 0.0099 | 0.0083 | 9.10% | 0.0087 | 0.0084 | 0.0084 | 0.0089 | 0.0086 | 0.0086 | 0.0088 | 0.0077 | 10.75% | 0.0093 | 0.0090 | 0.0090 | 0.0086 | 0.0094 | 0.0092 | 0.0096 | 0.0084 | 9.34% |
| | 24 | 0.1429 | 0.1397 | 0.1410 | 0.1088 | 0.1409 | 0.1430 | 0.1411 | 0.1145 | 19.85% | 0.1231 | 0.1190 | 0.1201 | 0.1109 | 0.1264 | 0.1191 | 0.1266 | 0.0972 | 21.05% | 0.1215 | 0.1184 | 0.1192 | 0.1085 | 0.1273 | 0.1212 | 0.1274 | 0.0947 | 22.09% |
| | 48 | 0.2076 | 0.2039 | 0.2054 | 0.1776 | 0.2032 | 0.2075 | 0.2033 | 0.1729 | 16.73% | 0.2393 | 0.2293 | 0.2346 | 0.1828 | 0.2300 | 0.2350 | 0.2302 | 0.1771 | 25.99% | 0.1678 | 0.1653 | 0.1667 | 0.1804 | 0.1708 | 0.1680 | 0.1709 | 0.1472 | 12.30% |
| traffic | 1 | 0.2228 | 0.2194 | 0.2200 | 0.2985 | 0.2191 | 0.2225 | 0.2232 | 0.2116 | 5.01% | 0.2132 | 0.2097 | 0.2108 | 0.2992 | 0.2103 | 0.2121 | 0.2133 | 0.2003 | 6.05% | 0.2151 | 0.2115 | 0.2122 | 0.3001 | 0.2128 | 0.2151 | 0.2160 | 0.2006 | 6.72% |
| | 24 | 0.4115 | 0.4086 | 0.4109 | 0.4181 | 0.4111 | 0.4115 | 0.4115 | 0.3859 | 6.20% | 0.4188 | 0.4143 | 0.4169 | 0.4173 | 0.4179 | 0.4181 | 0.4185 | 0.3820 | 8.80% | 0.4269 | 0.4248 | 0.4258 | 0.4179 | 0.4266 | 0.4269 | 0.4272 | 0.3986 | 6.63% |
| | 48 | 0.4475 | 0.4439 | 0.4477 | 0.4528 | 0.4495 | 0.4479 | 0.4497 | 0.4123 | 7.85% | 0.4328 | 0.4298 | 0.4324 | 0.4462 | 0.4329 | 0.4316 | 0.4331 | 0.4120 | 4.82% | 0.4566 | 0.4532 | 0.4570 | 0.4485 | 0.4566 | 0.4572 | 0.4568 | 0.4265 | 6.58% |
| weather | 1 | 0.0572 | 0.0568 | 0.0560 | 0.0582 | 0.0569 | 0.0573 | 0.0575 | 0.0510 | 10.95% | 0.0547 | 0.0542 | 0.0540 | 0.0596 | 0.0565 | 0.0577 | 0.0579 | 0.0505 | 7.67% | 0.0625 | 0.0617 | 0.0610 | 0.0560 | 0.0613 | 0.0620 | 0.0616 | 0.0521 | 16.69% |
| | 24 | 0.1780 | 0.1760 | 0.1776 | 0.2050 | 0.1820 | 0.1777 | 0.1820 | 0.1722 | 3.23% | 0.1878 | 0.1847 | 0.1868 | 0.2082 | 0.1816 | 0.1774 | 0.1819 | 0.1730 | 7.92% | 0.1733 | 0.1719 | 0.1733 | 0.2105 | 0.1743 | 0.1726 | 0.1743 | 0.1700 | 1.90% |
| | 48 | 0.2223 | 0.2207 | 0.2223 | 0.2629 | 0.2240 | 0.2218 | 0.2240 | 0.2170 | 2.39% | 0.2225 | 0.2204 | 0.2224 | 0.2632 | 0.2239 | 0.2211 | 0.2240 | 0.2147 | 3.49% | 0.2235 | 0.2213 | 0.2230 | 0.2623 | 0.2283 | 0.2236 | 0.2283 | 0.2170 | 2.89% |

# G  METHOD EFFICIENCY

We selected two datasets (ETTm1, traffic) to evaluate the efficiency. We summarized the GPU memory usage and deployment time required by various online methods during online deployment. The reason for choosing these two datasets is that the first dataset has a small dimensionality of only 7, while the second dataset has a large dimensionality of 861. Therefore, these two datasets represent significantly different scenarios. The experiments were conducted on a computer with an RTX 4090D and an i5-12400F. The results are reported in Figure 7.

On ETTm1 dataset, the difference in memory usage among the methods is not significant, with all requiring around 450MB. On the traffic dataset, some methods, such as DSOF and OGD, need over 7GB of memory, while our method only requires about 4.5GB. However, in terms of runtime, our method is not as fast as most baseline methods, but the additional time required is still acceptable.

Finally, we provide a theoretical analysis of the computational cost of our method. Our approach has both advantage and disadvantage.

The advantage is that since we extract features for subsequent operations, the feature encoder does not require gradient computation. This significantly reduces memory and computational costs. In contrast, methods like DSOF and OGD need to update all model parameters, which highlights the strength of our method. The disadvantage is that we need to compute historical gradients based on a batch of data, while other online prediction methods only process single samples. This introduces additional memory and computational overhead. Overall, since our method avoids gradient computation for most parts of the prediction model, the advantage could outweigh the disadvantage—especially for larger models.

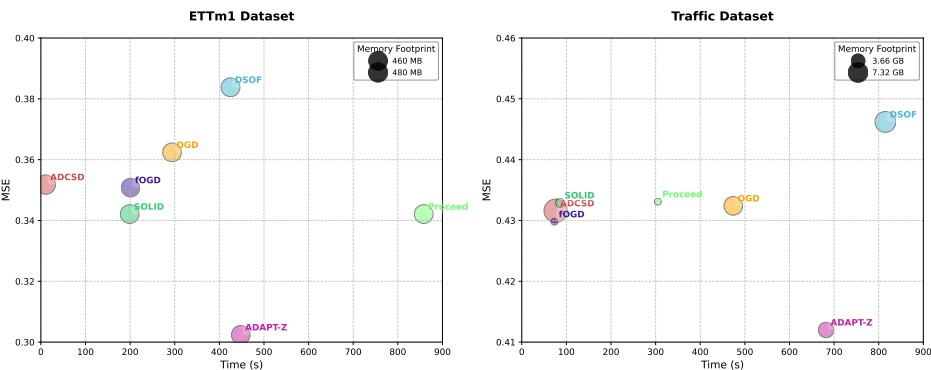

Figure 7: Method efficiency comparison (Base model is iTransformer)

# H  COMPARISON WITH TEST TIME ADAPTION METHODS

Additionally, there are some recent works about test time adaptation for time series forecasting, such as PETSA (Medeiros et al., 2025) and TAFAS (Kim et al., 2025). Although these studies do not explicitly name their tasks as online time series forecasting, we find their focus highly similar to ours—both involve adapting a pre-trained time series forecasting model during deployment to handle distribution shifts. Therefore, we compared these methods using iTransformer and SOFTS on several datasets, and the results are shown in Table 10. It can be seen that ADAPT-Z still achieves the best prediction accuracy in most cases compared to these methods.

# I  VISUALIZATION OF ONLINE PREDICTION RESULTS

We plot several figures showing the true values and predicted values for the last 100 time steps in the first dimension under different prediction steps. The blue, yellow, and green lines in these figures represent the true values, the directly deployed predicted values, and the predicted values

Table 10: The results of test time adaption methods

| Base model | | iTransformer | | | | SOFTS | | | |
|---|---|---|---|---|---|---|---|---|---|
| Dataset | H | Ori | TAFAS | PETSA | ADAPT-Z | Ori | TAFAS | PETSA | ADAPT-Z |
| ETTh1 | 1 | 0.1228 | 0.1187 | 0.1183 | **0.1127** | 0.1211 | 0.1152 | 0.1159 | **0.1119** |
| | 24 | 0.3325 | 0.3146 | 0.3306 | **0.3126** | 0.3201 | 0.3069 | 0.3090 | **0.3031** |
| | 48 | 0.3714 | 0.3598 | **0.3596** | 0.3626 | 0.3632 | 0.3521 | **0.3519** | 0.3584 |
| ETTm1 | 1 | 0.0570 | 0.0566 | 0.0568 | **0.0566** | 0.0564 | 0.0545 | 0.0544 | **0.0528** |
| | 24 | 0.2702 | 0.2565 | 0.2572 | **0.2293** | 0.2405 | 0.2221 | 0.2246 | **0.2166** |
| | 48 | 0.3655 | 0.3334 | 0.3331 | **0.3024** | 0.3126 | 0.2959 | 0.2962 | **0.2877** |
| ETTh2 | 1 | 0.0696 | 0.0687 | 0.0686 | **0.0674** | 0.0699 | 0.0693 | 0.0693 | **0.0659** |
| | 24 | 0.1827 | 0.1787 | 0.1782 | **0.1771** | 0.1796 | 0.1769 | 0.1761 | **0.1714** |
| | 48 | 0.2382 | 0.2337 | 0.2310 | **0.2302** | 0.2306 | 0.2279 | 0.2252 | **0.2239** |
| ETTm2 | 1 | 0.0327 | 0.0326 | 0.0327 | **0.0318** | 0.0335 | 0.0334 | 0.0333 | **0.0319** |
| | 24 | 0.1134 | 0.1132 | 0.1134 | **0.1082** | 0.1073 | 0.1068 | 0.1061 | **0.1037** |
| | 48 | 0.1449 | 0.1433 | 0.1435 | **0.1423** | 0.1430 | 0.1416 | 0.1417 | **0.1403** |
| weather | 1 | 0.0547 | 0.0536 | 0.0534 | **0.0505** | 0.0591 | 0.0583 | 0.0585 | **0.0531** |
| | 24 | 0.1878 | 0.1813 | 0.1816 | **0.1730** | 0.1695 | **0.1647** | 0.1658 | 0.1675 |
| | 48 | 0.2225 | 0.2147 | 0.2184 | **0.2147** | 0.2426 | 0.2315 | 0.2379 | **0.2155** |
| traffic | 1 | 0.2132 | 0.2055 | 0.2087 | **0.2003** | 0.2009 | 0.1969 | 0.1990 | **0.1959** |
| | 24 | 0.4188 | 0.4001 | 0.4055 | **0.3820** | 0.3559 | 0.3520 | 0.3540 | **0.3520** |
| | 48 | 0.4328 | 0.4220 | 0.4257 | **0.4120** | 0.3879 | 0.3856 | 0.3851 | **0.3848** |

using ADAPT-Z online deployment, respectively. The base model used in plotting these figures is SOFTS.

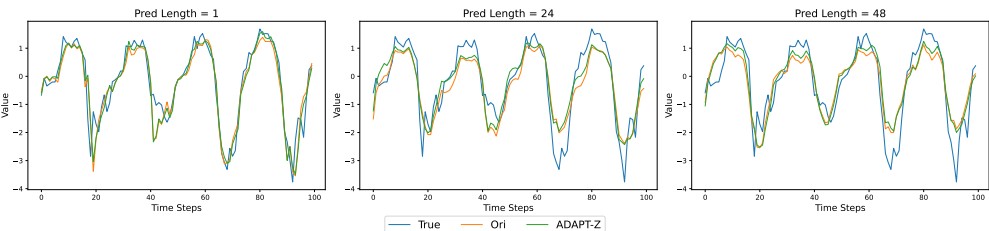

Figure 8: Visualization of ETTh1 dataset

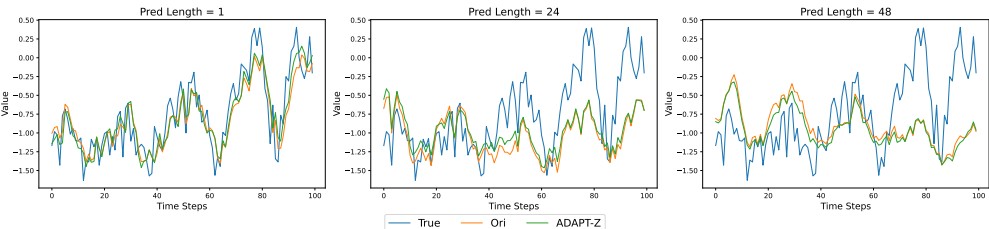

Figure 9: Visualization of ETTh2 dataset

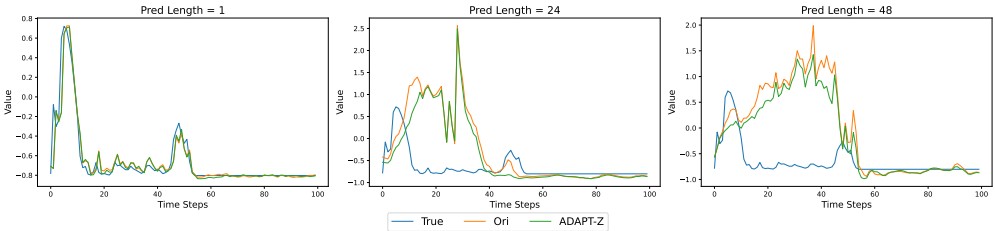

Figure 10: Visualization of solar dataset

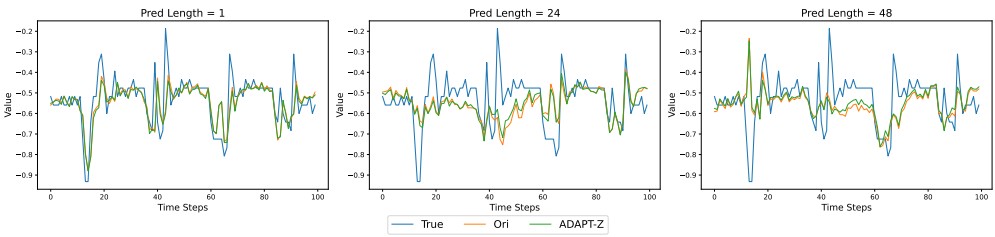

Figure 11: Visualization of electricity dataset

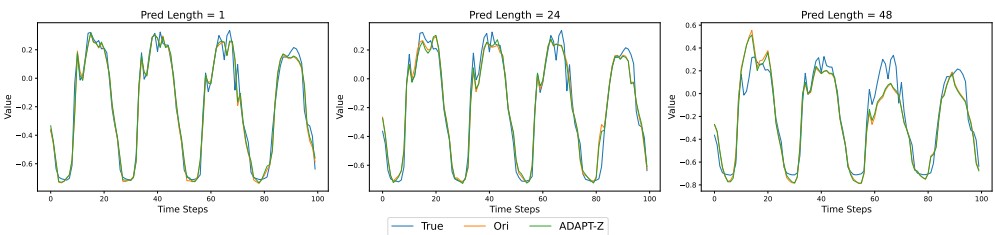

Figure 12: Visualization of traffic dataset

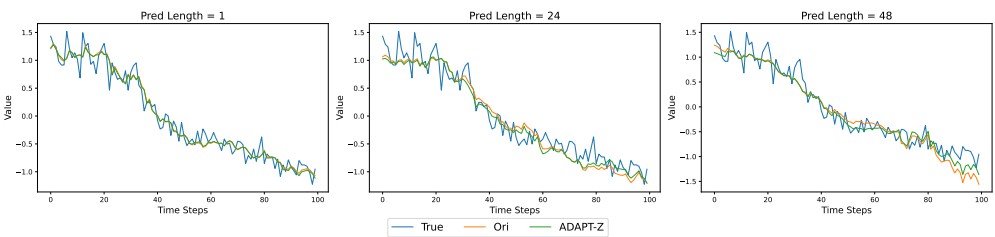

Figure 13: Visualization of PEMS03 dataset

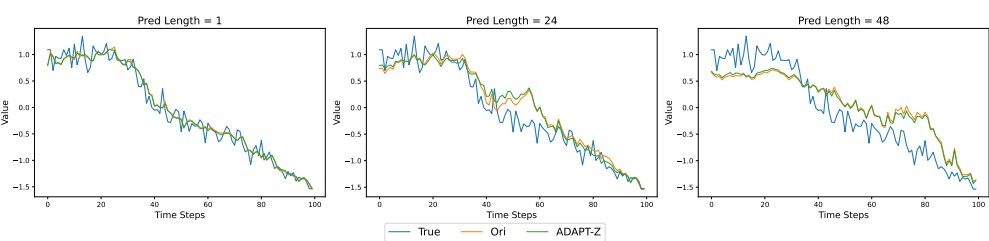

Figure 14: Visualization of PEMS08 dataset

