# OpenReview forum: "Online time series prediction using feature adjustment"
_ICLR.cc/2026/Conference — ICLR 2026 Poster_

### Official Review · Reviewer_a4GN · 2025-10-30

**Soundness:** 2
**Presentation:** 2
**Contribution:** 2
**Rating:** 4
**Confidence:** 4

**Summary:**

This paper tackles the challenge of distribution shift in online time series forecasting. It introduces a paradigm that deviates from conventional methods of updating model parameters. The authors posit that distribution shifts are fundamentally caused by changes in underlying latent factors, and therefore, it is more effective to directly adjust the feature representations produced by a model's encoder.
To this end, they propose ADAPT-Z (Automatic Delta Adjustment via Persistent Tracking in Z-space), a method that uses a lightweight adapter module to predict a corrective "delta" for the feature vector.

**Strengths:**

The paper is built upon a clear and intuitive premise. And the proposed method is tested on multiple public datasets using different base forecasting models, and its performance is compared against several baseline approaches. The authors also include supplementary analyses, such as ablation studies and a discussion of hyperparameters, which offer some insight into the behavior of the proposed method.

**Weaknesses:**

1. **"Historical Gradient"**: The method relies on a "historical gradient" computed from a batch of data ending at timestep t-k to inform the feature correction at the current timestep t. While this is a pragmatic approach to handling delayed feedback, this term is not a true gradient for the current step but rather a delayed and averaged approximation. The paper could provide a clearer discussion on the implications of this temporal mismatch, especially in scenarios with abrupt, non-gradual shifts where this historical information might be stale or misleading.

2. **Insufficient Engagement with Test-Time Adaptation (TTA) Literature:** The task of adapting a pre-trained model to a stream of incoming data under distribution shift is the central problem in the Test-Time Adaptation (TTA) field. The paper does not adequately position its work within this highly relevant body of literature in the main text. A more thorough comparison and discussion of the similarities and differences with TTA methods would provide better context for the paper's contributions.

3. **Practical Concerns Regarding Computational Efficiency:** The efficiency analysis in the appendix reveals that ADAPT-Z, while often more memory-efficient than full-model fine-tuning, is slower in terms of runtime than most baseline methods. For real-world online applications where prediction latency is a critical constraint, this added computational cost could be a significant drawback. This trade-off between accuracy and speed deserves a more prominent discussion in the main paper.

4. **Potential Ambiguity in Dataset Split Justification:** The authors justify their use of a 60/10/30 train/val/test split by arguing it is more "realistic" than the 25/5/70 splits used in some prior work. While their reasoning is plausible, this choice results in a significantly shorter online deployment period for evaluation. This could make the adaptation challenge less severe compared to enduring a distribution shift over 70% of the data, thereby complicating direct comparisons with results from papers that used the longer test split.

**Questions:**

Please see the weaknesses.

---

> ### Author Response · Authors · 2025-11-20
> **Response to Reviewer a4GN (Part 1)**
>
> ## Q1: Delayed gradient
> Thank you for the feedback. You raised a valid point. Although we use gradients from the t-k step, it is important to note that the prediction at t-k step involves the true value at time t. Since we use the true value at time t to calculate the gradients, this means our update at step t actually incorporates the latest available data. Therefore, even though the gradients are delayed, their impact may not be overly significant.
>
> Regarding abrupt, non-gradual shifts, please refer to our response to Reviewer 3's fourth question. We simulated these shifts by artificially modifying test set data and conducted experiments across multiple datasets. ADAPT-Z still demonstrated competitive performance, suggesting that our method can adapt relatively well even with delayed gradient feedback. Finally, how to handle delayed feedback remains an important direction for future research.
>
> ## Q2: Comparison with TTA methods
> Thank you for your comment! We have reviewed a portion of the Test-Time Adaptation (TTA) literature and found that this technique was initially developed primarily for classification tasks. Classic methods include updating parameters in normalization layers, performing data augmentation, using mean teachers and memory banks, etc. However, these methods may not be applicable directly to time series data.
>
> Several existing TTA works for time series, such as ADCSD, PETSA, and TAFAS, are discussed in our paper. Specifically, they all align with the two challenges we outlined in the introduction: 1) deciding which parameters to update and 2) determining how to update them.
>
> For the "which parameters" question, ADCSD updates an adapter to correct model's outputs, while PETSA and TAFAS use adapters to modify the model's inputs and outputs. PETSA even employs a LoRA-like structure for its adapter. Regarding the "how to update" question, ADCSD performs adaptation by online gradient descent. PETSA and TAFAS utilize specific update schemes, such as using partially-observed ground truth for updates.
>
> According to our conceptual framework, distribution shifts stem from changes in the latent variables governing the data distribution. Therefore, updating these latent representations is more effective than updating model parameters. Consequently, for the "what to update" problem, we propose updating the feature representations. For the "how to update" problem, we use an adapter that integrates current features and historical gradient information to perform the updates.
>
> We hope this provides a thorough comparison and discussion of the similarities and differences with existing TTA methods. And we will add a subsection in the Appendix A of our paper to discuss these TTA methods.
>
> **Reference**
>
> Guo, Pengxin, et al. "Online Test-Time Adaptation of Spatial–Temporal Traffic Flow Forecasting." IEEE Transactions on Intelligent Transportation Systems (2025). (ADCSD)
>
> Medeiros, Heitor R., et al. "Accurate Parameter-Efficient Test-Time Adaptation for Time Series Forecasting." arXiv preprint arXiv:2506.23424 (2025). (PETSA)
>
> Kim, HyunGi, et al. "Battling the non-stationarity in time series forecasting via test-time adaptation." Proceedings of the AAAI Conference on Artificial Intelligence. Vol. 39. No. 17. 2025. (TAFAS)
>
> Wang, Zixin, et al. "In search of lost online test-time adaptation: A survey." International Journal of Computer Vision 133.3 (2025): 1106-1139.

---

> ### Author Response · Authors · 2025-11-20
> **Response to Reviewer a4GN (Part 2)**
>
> ## Q3: Computational efficiency
> Thank you for your feedback. While our method does need a longer runtime compared to some baseline approaches, the overall computational cost remains practical. Even on a high-dimensional dataset like Traffic (with 862 dimensions), our method can complete testing on the entire test set in approximately 10 minutes. Considering the test set contains over 3,000 time steps, this means our method can process about 5 time steps per second. For lower-dimensional datasets like ETTm2 (7 dimensions), it can process several dozen time steps per second. This level of computational efficiency should be sufficient for real-world deployment scenarios.
>
> ## Q4: Dataset splitting
> Thank you for your question! We have two main reasons for not using the 25/5/70 data split. First, as mentioned in our paper, this would result in an excessively long online deployment phase (sometimes nearly two years). In practice, it is unlikely for a model to be deployed for so long with only online adaptation and no retraining.
> Additionally, although not stated in the paper, using only 25\% of the data for training cannot guarantee that the model is fully trained. This creates an ambiguity: if the prediction accuracy improves during deployment, it is difficult to determine whether the improvement comes from addressing distribution shift, or simply from the model being trained with more data and thus reaching its potential.
> Regarding this issue of insufficient model training, please refer to the experimental results in the DSOF paper (which uses a 25/5/75 split). On datasets like ETTm1 and Weather, more advanced models like iTransformer, PatchTST, and NSTransformer sometimes underperform the simple linear model DLinear. This counter-intuitive result is likely because these complex models are not adequately trained with only 25\% of the data.
>
> **Reference**
>
> Lau, Ying-yee Ava, Zhiwen Shao, and Dit-Yan Yeung. "Fast and Slow Streams for Online Time Series Forecasting Without Information Leakage." The Thirteenth International Conference on Learning Representations. 2025. (DSOF)

---

### Official Review · Reviewer_tYvU · 2025-10-30

**Soundness:** 3
**Presentation:** 3
**Contribution:** 3
**Rating:** 6
**Confidence:** 5

**Summary:**

The paper introduces ADAPT-Z (Automatic Delta Adjustment via Persistent Tracking in Z-space), a novel online time-series forecasting framework that adapts to distribution shifts by updating feature representations instead of model parameters. It employs a lightweight adapter network that fuses current latent features and historical gradients to generate correction terms ($\delta_t$) in the feature space, effectively mitigating delayed feedback in multi-step forecasting. Extensive experiments on 13 datasets and three base models (iTransformer, SOFTS, TimesNet) demonstrate consistent gains over state-of-the-art online learning methods such as DSOF, SOLID, Proceed, and ADCSD. The paper highlights that feature-space adaptation offers a more stable and interpretable alternative to conventional parameter updates under non-stationary environments.

**Strengths:**

The paper offers an original perspective by focusing on feature-space adaptation rather than parameter updates, addressing distribution shifts through latent factor correction. It provides strong technical quality, supported by extensive experiments across 13 datasets and several architectures with consistent improvements. The methodology is clearly presented, including detailed pseudo-code, ablation studies, and reproducibility information. Its significance lies in showing that updating latent feature representations is a stable and general strategy for online time-series forecasting under non-stationary conditions.

**Weaknesses:**

While the experiments are extensive and demonstrate consistent gains, they lack statistical robustness, as no variance or confidence intervals are reported across different random seeds, making it difficult to assess the reliability of the improvements. The evaluation could be broadened to include a wider range of model architectures, such as MLP-based and linear forecasting models, which are common baselines in the time-series literature. The computational efficiency of the approach is also unclear, particularly the memory and runtime overhead introduced by storing and updating historical gradients, which may limit scalability in long-horizon or high-frequency settings. Furthermore, assessing performance under stronger distribution shifts or longer forecasting horizons would provide stronger evidence of generalizability and robustness across real-world deployment scenarios.

**Questions:**

1. Could the authors provide statistical analysis (e.g., variance or confidence intervals across multiple random seeds) to assess the consistency and significance of the reported improvements? P.S.: This can be for a small scenario just to show that we can improve with the proposed method. I am not asking for anything new or extensive here. It is just to make the paper stronger.

2. How well does ADAPT-Z generalize across different backbone families, such as linear or MLP-based forecasting models, which are widely used in the literature?

3. What is the computational and memory overhead associated with maintaining and updating historical gradients during online adaptation, and how does this scale with longer horizons or larger datasets?

4. Have the authors evaluated the method under stronger or abrupt distribution shifts, such as sudden regime changes, to test its robustness in more realistic online environments?

---

> ### Author Response · Authors · 2025-11-20
> **Response to Reviewer tYvU (Part 1)**
>
> ## Q1: Robustness across different random seeds
>
>  We report the experimental results for iTransformer using three random seeds in the Appendix (A.6.1). This demonstrates that our method remains effective across different random seeds. Regarding your interest in statistical robustness, we have conducted additional experiments on the SOFTS and TimesNET models using the ETT datasets with three random seeds (2024, 2025, and 2026). The results, including both the MSE and the standard deviation of MSE, are reported below.
>
> **SOFTS**
>
> | Dataset | H  | Ori_mean | Ori_std | fOGD_mean | fOGD_std | OGD_mean | OGD_std | DSOF_mean | DSOF_std | SOLID_mean | SOLID_std | ADCSD_mean | ADCSD_std | Proceed_mean | Proceed_std | ADAPT-Z_mean | ADAPT-Z_std |
> |---------|----|----------|---------|-----------|----------|----------|---------|-----------|----------|------------|-----------|------------|-----------|--------------|-------------|--------------|-------------|
> | **ETTh1** | 1  | 0.1211 | 0.0021 | 0.1209 | 0.0028 | 0.1192 | 0.0020 | 0.1316 | 0.0004 | 0.1194 | 0.0024 | 0.1199 | 0.0026 | 0.1203 | 0.0025 | **0.1122** | 0.0026 |
> |         | 24 | 0.3182 | 0.0012 | 0.3217 | 0.0020 | 0.3223 | 0.0010 | 0.3312 | 0.0010 | 0.3228 | 0.0031 | 0.3172 | 0.0011 | 0.3210 | 0.0031 | **0.3029** | 0.0013 |
> |         | 48 | 0.3666 | 0.0063 | 0.3656 | 0.0054 | 0.3657 | 0.0064 | 0.3894 | 0.0020 | 0.3600 | 0.0054 | 0.3618 | 0.0066 | 0.3625 | 0.0053 | **0.3522** | 0.0029 |
> | **ETTh2** | 1  | 0.0702 | 0.0023 | 0.0700 | 0.0024 | 0.0692 | 0.0024 | 0.0691 | 0.0018 | 0.0700 | 0.0019 | 0.0699 | 0.0026 | 0.0701 | 0.0020 | **0.0661** | 0.0017 |
> |         | 24 | 0.1801 | 0.0009 | 0.1791 | 0.0009 | 0.1788 | 0.0008 | 0.1834 | 0.0008 | 0.1797 | 0.0020 | 0.1807 | 0.0008 | 0.1784 | 0.0021 | **0.1729** | 0.0016 |
> |         | 48 | 0.2305 | 0.0031 | 0.2327 | 0.0031 | 0.2309 | 0.0033 | 0.2906 | 0.0079 | 0.2310 | 0.0035 | 0.2318 | 0.0032 | 0.2300 | 0.0035 | **0.2239** | 0.0016 |
> | **ETTm1** | 1  | 0.0570 | 0.0008 | 0.0554 | 0.0008 | 0.0552 | 0.0003 | 0.0741 | 0.0006 | 0.0563 | 0.0004 | 0.0555 | 0.0009 | 0.0566 | 0.0005 | **0.0523** | 0.0002 |
> |         | 24 | 0.2384 | 0.0038 | 0.2385 | 0.0050 | 0.2390 | 0.0042 | 0.2888 | 0.0051 | 0.2419 | 0.0032 | 0.2399 | 0.0040 | 0.2396 | 0.0033 | **0.2148** | 0.0012 |
> |         | 48 | 0.3154 | 0.0140 | 0.3081 | 0.0103 | 0.3122 | 0.0138 | 0.3935 | 0.0061 | 0.3156 | 0.0107 | 0.3087 | 0.0094 | 0.3123 | 0.0110 | **0.2865** | 0.0030 |
> | **ETTm2** | 1  | 0.0335 | 0.0005 | 0.0336 | 0.0005 | 0.0337 | 0.0005 | 0.0448 | 0.0001 | 0.0338 | 0.0006 | 0.0335 | 0.0005 | 0.0337 | 0.0005 | **0.0322** | 0.0002 |
> |         | 24 | 0.1062 | 0.0012 | 0.1078 | 0.0012 | 0.1075 | 0.0012 | 0.1532 | 0.0015 | 0.1079 | 0.0026 | 0.1074 | 0.0011 | 0.1072 | 0.0026 | **0.1032** | 0.0001 |
> |         | 48 | 0.1439 | 0.0017 | 0.1422 | 0.0016 | 0.1444 | 0.0016 | 0.2063 | 0.0086 | 0.1426 | 0.0017 | 0.1415 | 0.0016 | 0.1423 | 0.0018 | **0.1399** | 0.0007 |

---

> ### Author Response · Authors · 2025-11-20
> **Response to Reviewer tYvU (Part 2)**
>
> **TimesNet**
>
> | Dataset | H  | Ori_mean | Ori_std | fOGD_mean | fOGD_std | OGD_mean | OGD_std | DSOF_mean | DSOF_std | SOLID_mean | SOLID_std | ADCSD_mean | ADCSD_std | Proceed_mean | Proceed_std | ADAPT-Z_mean | ADAPT-Z_std |
> |---------|----|----------|---------|-----------|----------|----------|---------|-----------|----------|------------|-----------|------------|-----------|--------------|-------------|--------------|-------------|
> | **ETTh1** | 1  | 0.1371 | 0.0020 | 0.1362 | 0.0029 | 0.1370 | 0.0016 | 0.1620 | 0.0004 | 0.1366 | 0.0025 | 0.1366 | 0.0027 | 0.1368 | 0.0029 | **0.1290** | 0.0031 |
> |         | 24 | 0.3481 | 0.0012 | 0.3420 | 0.0020 | 0.3397 | 0.0009 | 0.4604 | 0.0010 | 0.3438 | 0.0034 | 0.3474 | 0.0013 | 0.3459 | 0.0035 | **0.3360** | 0.0015 |
> |         | 48 | 0.3954 | 0.0070 | 0.3854 | 0.0058 | 0.3836 | 0.0054 | 0.5793 | 0.0018 | 0.3904 | 0.0046 | 0.3950 | 0.0056 | 0.3929 | 0.0056 | **0.3763** | 0.0032 |
> | **ETTh2** | 1  | 0.0775 | 0.0031 | 0.0775 | 0.0025 | 0.0763 | 0.0028 | 0.0781 | 0.0016 | 0.0787 | 0.0020 | 0.0765 | 0.0027 | 0.0776 | 0.0023 | **0.0750** | 0.0018 |
> |         | 24 | 0.1984 | 0.0009 | 0.1956 | 0.0008 | 0.1965 | 0.0008 | 0.2623 | 0.0007 | 0.1983 | 0.0026 | 0.1982 | 0.0007 | 0.2025 | 0.0025 | **0.1909** | 0.0014 |
> |         | 48 | 0.2428 | 0.0026 | 0.2340 | 0.0027 | 0.2424 | 0.0028 | 0.3461 | 0.0076 | 0.2410 | 0.0027 | 0.2398 | 0.0032 | 0.2362 | 0.0033 | **0.2402** | 0.0021 |
> | **ETTm1** | 1  | 0.0585 | 0.0010 | 0.0594 | 0.0009 | 0.0572 | 0.0004 | 0.0766 | 0.0006 | 0.0576 | 0.0005 | 0.0582 | 0.0008 | 0.0581 | 0.0004 | **0.0548** | 0.0002 |
> |         | 24 | 0.2675 | 0.0047 | 0.2715 | 0.0047 | 0.2595 | 0.0042 | 0.3494 | 0.0067 | 0.2761 | 0.0040 | 0.2631 | 0.0050 | 0.2668 | 0.0033 | **0.2425** | 0.0013 |
> |         | 48 | 0.3625 | 0.0133 | 0.3619 | 0.0130 | 0.3554 | 0.0158 | 0.4351 | 0.0057 | 0.3580 | 0.0129 | 0.3492 | 0.0085 | 0.3614 | 0.0123 | **0.3073** | 0.0029 |
> | **ETTm2** | 1  | 0.0329 | 0.0006 | 0.0328 | 0.0005 | 0.0328 | 0.0004 | 0.0439 | 0.0000 | 0.0333 | 0.0005 | 0.0335 | 0.0006 | 0.0336 | 0.0006 | **0.0326** | 0.0002 |
> |         | 24 | 0.1128 | 0.0016 | 0.1151 | 0.0014 | 0.1126 | 0.0010 | 0.1673 | 0.0015 | 0.1148 | 0.0031 | 0.1141 | 0.0012 | 0.1139 | 0.0023 | **0.1054** | 0.0001 |
> |         | 48 | 0.1560 | 0.0020 | 0.1509 | 0.0012 | 0.1558 | 0.0021 | 0.2434 | 0.0099 | 0.1511 | 0.0018 | 0.1535 | 0.0016 | 0.1523 | 0.0020 | **0.1438** | 0.0008 |

---

> ### Author Response · Authors · 2025-11-20
> **Response to Reviewer tYvU (Part 3)**
>
> ## Q2: More base model
> Thank you for your comment. We would like to point out that SOFTS is indeed an MLP-based model. As for linear models, we did not include them in our experiments because a simple linear model cannot be decomposed into an encoder and a prediction head. Additionally, if you have further concerns about the generality of our method, please refer to our response to Reviewer 1's fourth question. We have also conducted additional experiments with TimeFilter and TimeKAN. These results further demonstrate the effectiveness of our approach.
>
> ## Q3: Memory and runtime overhead
> Thank you for your question. In fact, our method only requires the historical gradients from the previous k steps when performing an update. Gradients from earlier steps do not need to be stored and can be discarded. Therefore, storing historical gradients does not require significant memory.
>
> Regarding computational time, we report the results in Appendix A.7. Our method completes online testing in minutes, even for datasets comprising thousands of time steps, corresponding to a processing rate of dozens of time steps per second. This demonstrates its suitability for practical deployment.
>
> The table below reports the number of time steps processed per second and the GPU memory consumption when running our algorithm on two datasets, Traffic and ETTm1, under forecast horizons of 96, 192, 336, and 720 steps. Notably, Traffic contains 862 dimensions, making it one of the largest datasets in our experiments.
>
> |         |       H      |  96  |  192 |  336 |  720 |
> |:-------:|:------------:|:----:|:----:|:----:|:----:|
> | Traffic | speed(itr/s) |   5  |   5  |   5  |   5  |
> |         |  memory(GB)  | 1.44 | 1.45 | 1.46 | 1.47 |
> |  ETTm1  | speed(itr/s) |  38  |  38  |  37  |  37  |
> |         |  memory(GB)  | 0.45 | 0.45 | 0.45 | 0.46 |
>
> As shown, the additional memory overhead and the decrease in iteration speed are negligible as the forecast horizon increases. This can be explained by analyzing the source of the overhead. The only structural difference between models predicting different horizons lies in the output dimension of the final linear layer. Consequently, even as the prediction horizon expands, the shape of the historical gradients that need to be stored remains unchanged because the parameter dimensions of the intermediate layers are fixed. Therefore, the additional memory cost is solely due to the slightly larger final linear layer. The additional computational overhead follows the same logic. For these reasons, the increased demand for memory and the  slowdown in processing speed are minimal as the prediction horizon grows.
>
> **Reference**
>
> Han, Lu, et al. "Softs: Efficient multivariate time series forecasting with series-core fusion." Advances in Neural Information Processing Systems 37 (2024): 64145-64175.

---

> ### Author Response · Authors · 2025-11-20
> **Response to Reviewer tYvU (Part 4)**
>
> ## Q4.1: Performance under stronger distribution to provide evidence of generalizability and robustness across real-world deployment scenarios
>
> Most existing papers on online time series forecasting conduct experiments on the datasets we mentioned, such as DSOF and SOLID. Compared to previous methods, we have increased the number of datasets used in our experiments. We now use a total of 12 datasets, all of which are real-world datasets. This should demonstrate generalizability and robustness across real-world deployment scenarios.
>
> To evaluate performance under more significant distribution shifts, we modified the test set data. The original data point at time t, denoted as $x_t$, was altered to $x_t + (t/L)sin(2π*t/T) + N(t/L,0.1)$, where $T$ represents the dataset's period and $L$ is the total length of the test dataset. This modification simulates a gradually intensifying distribution shift over time. We conducted additional experiments using the SOFTS model and the ETT datasets.
>
> | Dataset |  H |   Ori   |   fOGD  |   OGD   |   DSOF  |  SOILD  |  ADCSD  | Proceed |   ADAPT-Z   |   IMP  |
> |:-------:|:--:|:-------:|:-------:|:-------:|:-------:|:-------:|:-------:|:-------:|:-----------:|:------:|
> |  ETTh1  |  1 | 0.1671  | 0.1482  | 0.1442  | 0.1584  | 0.1531  | 0.1535  | 0.1395  | **0.1306** | 21.87% |
> |         | 24 | 0.4663  | 0.3764  | 0.4831  | 0.3874  | 0.3984  | 0.3982  | 0.3820  | **0.3415** | 26.76% |
> |         | 48 | 0.4697  | 0.4090  | 0.4345  | 0.4283  | 0.4294  | 0.4248  | 0.4123  | **0.3960** | 15.70% |
> |  ETTh2  |  1 | 0.0886  | 0.0856  | 0.0864  | 0.0858  | 0.0856  | 0.0877  | 0.0857  | **0.0832** |  6.08% |
> |         | 24 | 0.2019  | 0.1903  | 0.1973  | 0.1980  | 0.1963  | 0.1933  | 0.1952  | **0.1883** |  6.74% |
> |         | 48 | 0.2695  | 0.2535  | 0.2573  | 0.2604  | 0.2583  | 0.2589  | 0.2533  | **0.2490** |  7.60% |
> |  ETTm1  |  1 | 0.0753  | 0.0724  | 0.0735  | 0.0732  | 0.0741  | 0.0750  | 0.0721  | **0.0669** | 11.09% |
> |         | 24 | 0.4340  | 0.3471  | 0.3625  | 0.3982  | 0.2983  | 0.3340  | 0.2891  | **0.2431** | 43.99% |
> |         | 48 | 0.5302  | 0.3674  | 0.4093  | 0.4391  | 0.3821  | 0.4482  | 0.3701  | **0.3125** | 41.05% |
> |  ETTm2  |  1 | 0.0507  | 0.4804  | 0.4824  | 0.4810  | 0.4803  | 0.4862  | 0.4814  | **0.0471** |  7.20% |
> |         | 24 | 0.1501  | 0.1341  | 0.1340  | 0.1472  | 0.1408  | 0.1460  | 0.1331  | **0.1248** | 16.85% |
> |         | 48 | 0.1852  | 0.1673  | 0.1662  | 0.1764  | 0.1609  | 0.1773  | 0.1678  | **0.1590** | 14.14% |
>
> It can be observed that our method maintains an advantage even when facing significant distribution shifts.
> Additionally, we considered applying the same modification only to the middle one-third of the test dataset. This simulates a scenario with two abrupt change points.  The results are are reported below.
>
> | Dataset |  H |   Ori   |   fOGD  |   OGD   |   DSOF  |  SOILD  |  ADCSD  | Proceed |   ADAPT-Z   |   IMP  |
> |:-------:|:--:|:-------:|:-------:|:-------:|:-------:|:-------:|:-------:|:-------:|:-----------:|:------:|
> |  ETTh1  |  1 | 0.1336  | 0.1193  | 0.1204  | 0.1301  | 0.1220  | 0.1310  | 0.1233  | **0.1193** | 10.70% |
> |         | 24 | 0.3647  | 0.3403  | 0.3448  | 0.3648  | 0.3340  | 0.3503  | 0.3391  | **0.3281** | 10.04% |
> |         | 48 | 0.3999  | 0.3867  | 0.3908  | 0.3991  | 0.3841  | 0.3974  | 0.3870  | **0.3806** |  4.84% |
> |  ETTh2  |  1 | 0.0765  | 0.0733  | 0.0739  | 0.0756  | 0.0746  | 0.0740  | 0.0735  | **0.0726** |  5.06% |
> |         | 24 | 0.1855  | 0.1804  | 0.1835  | 0.1855  | 0.1824  | 0.1857  | 0.1833  | **0.1789** |  3.54% |
> |         | 48 | 0.2405  | 0.2400  | 0.2482  | 0.2483  | 0.2389  | 0.2411  | 0.2388  | **0.2344** |  2.54% |
> |  ETTm1  |  1 | 0.0624  | 0.0592  | 0.0611  | 0.0601  | 0.0611  | 0.0612  | 0.0598  | **0.0578** |  7.32% |
> |         | 24 | 0.2904  | 0.2472  | 0.2583  | 0.2514  | 0.2493  | 0.2753  | 0.2471  | **0.2299** | 20.83% |
> |         | 48 | 0.3700  | 0.3308  | 0.3374  | 0.3371  | 0.3415  | 0.3572  | 0.3201  | **0.3025** | 18.25% |
> |  ETTm2  |  1 | 0.0395  | 0.3801  | 0.3811  | 0.3914  | 0.3805  | 0.3944  | 0.3799  | **0.0375** |  5.00% |
> |         | 24 | 0.1192  | 0.1163  | 0.1183  | 0.1170  | 0.1152  | 0.1195  | 0.1162  | **0.1141** |  4.23% |
> |         | 48 | 0.1555  | 0.1524  | 0.1552  | 0.1562  | 0.1533  | 0.1550  | 0.1522  | **0.1498** |  3.68% |
>
> ## Q4.2: Longer forecasting horizons
>
> Regarding longer prediction horizons, please refer to our response to the final question from Reviewer 2. We conducted additional experiments using the ETT datasets with prediction horizons of 96, 192, 336, and 720 steps. The results consistently demonstrate the effectiveness of our method.

---

### Official Review · Reviewer_Twn1 · 2025-10-31

**Soundness:** 2
**Presentation:** 2
**Contribution:** 2
**Rating:** 4
**Confidence:** 4

**Summary:**

This paper proposes ADAPT-Z for online time series forecasting tasks to address distribution shift issues, particularly the critical problem of delayed feedback in multi-step forecasting. The paper argues that distribution shifts stem from changes in the underlying factors influencing the data, and ADAPT-Z introduces an adapter module that leverages current feature combined with historical gradient information to achieve high-quality parameter updates even in the presence of delays.

**Strengths:**

1. ADAPT-Z focuses on features to address shift, dynamically estimating bias using features combined with historical gradient information, while also exhibiting strong compatibility and can be used alongside normalization methods.
2. The experiments effectively corroborate most of the claims made in the paper.
3. The experiments are comprehensive, having investigated most aspects of the research that warrant further exploration.
4. The workflow of this paper is very clear.

**Weaknesses:**

1. The paper identifies concept drift as a key limitation of existing research schemes, but the method does not explain how ADAPT-Z addresses it, making it unclear why concept drift is mentioned.
2. For the main results, the metric used for evaluation is MSE. Could the performance of various methods also be examined using MAE?
3. In section 4.3.3, the paper explores the performance impact of extracting feature from different regions of iTransformer. Could this exploration be extended to feature locations in other models? For instance, the models used in this paper, such as SOFTS and TimesNet. Based on the best-performing feature locations across several models, could any commonalities be identified?
4. In section 4.3.6, the paper states: "our shorter prediction horizons (1-48 steps) experience more volatile distribution shifts. This instability makes it challenging for normalization method to consistently align input-output distributions across diverse time segments." Could the stable improvement of combining ADAPT-Z with normalization methods be explored under longer prediction horizons? Simultaneously, could ADAPT-Z's leading performance in forecasting tasks be examined under these longer prediction horizons as well?
5. The idea is quite simple. Can you clarify the novelty lies in the proposed method?

**Questions:**

As in Weaknesses

---

> ### Author Response · Authors · 2025-11-20
> **Response to Reviewer Twn1 (Part 1)**
>
> ## Q1: Concept drift
> Thank you for your comment. ADAPT-Z implicitly handles concept drift through feature-space correction. Its core mechanism works as follows: The prediction model is divided into an encoder (f) and a prediction head (g). The encoder extracts feature representations z from observed data, while the prediction head uses z for forecasting. Due to model obsolescence, the prediction g(z) often differs from the true value y. Therefore, ADAPT-Z finds a correction term $\delta$ such that g(z+$\delta$) approximates y. This approach essentially adjusts feature representations to adapt to changes in the current data distribution, thereby mitigating concept drift. Although the term "concept drift" is not explicitly used in the method description, ADAPT-Z's entire design aims to handle distribution shifts, including concept drift. It keeps the model up-to-date through feature adjustment and online learning, enabling it to adapt to distribution changes.
>
> ## Q2 MAE
> Thank you for your feedback.
> Our method also outperforms existing approaches on the MAE metric. The reason we did not include MAE in the main text was to keep the table manageable. We report the  MAE on the ETT datasets in the table below (The reported MAE is the average value for three prediction models).
>
> | Dataset | H  | Ori    | fOGD   | OGD    | DSOF   | SOILD  | ADCSD  | Proceed | ADAPT-Z |
> |---------|----|--------|--------|--------|--------|--------|--------|---------|---------|
> | **ETTh1** | 1  | 0.2357 | 0.2336 | 0.2326 | 0.2458 | 0.2346 | 0.2346 | 0.2346  | **0.2262** |
> |         | 24 | 0.3856 | 0.3835 | 0.3845 | 0.3968 | 0.3847 | 0.3853 | 0.3846  | **0.3725** |
> |         | 48 | 0.4162 | 0.4141 | 0.4151 | 0.4274 | 0.4161 | 0.4151 | 0.4153  | **0.4037** |
> | **ETTh2** | 1  | 0.1824 | 0.1805 | 0.1795 | 0.1887 | 0.1805 | 0.1824 | 0.1817  | **0.1735** |
> |         | 24 | 0.2845 | 0.2825 | 0.2825 | 0.2917 | 0.2844 | 0.2837 | 0.2840  | **0.2785** |
> |         | 48 | 0.3239 | 0.3223 | 0.3231 | 0.3315 | 0.3228 | 0.3223 | 0.3213  | **0.3133** |
> | **ETTm1** | 1  | 0.1513 | 0.1499 | 0.1489 | 0.1581 | 0.1506 | 0.1503 | 0.1499  | **0.1430** |
> |         | 24 | 0.3158 | 0.3142 | 0.3142 | 0.3244 | 0.3143 | 0.3142 | 0.3149  | **0.3055** |
> |         | 48 | 0.3959 | 0.3937 | 0.3953 | 0.4039 | 0.3949 | 0.3937 | 0.3947  | **0.3698** |
> | **ETTm2** | 1  | 0.1135 | 0.1122 | 0.1153 | 0.1204 | 0.1183 | 0.1163 | 0.1153  | **0.1079** |
> |         | 24 | 0.2222 | 0.2203 | 0.2213 | 0.2295 | 0.2220 | 0.2203 | 0.2213  | **0.2169** |
> |         | 48 | 0.2516 | 0.2480 | 0.2512 | 0.3000 | 0.2515 | 0.2508 | 0.2564  | **0.2473** |

---

> ### Author Response · Authors · 2025-11-20
> **Response to Reviewer Twn1 (Part 2)**
>
> ## Q3: Feature location
> We also conducted experiments on the SOFTS model. Similar to the iTransformer experiments, we extracted features from various layers for feature correction. The final results are shown in the table below.
>
> | Dataset | electricity_1 | electricity_24 | electricity_48 | PEMS03_1 | PEMS03_24 | PEMS03_48 | PEMS07_1 | PEMS07_24 | PEMS07_48 | solar_1 | solar_24 | solar_48 | weather_1 | weather_24 | weather_48 |
> |--------|---------------|----------------|----------------|----------|-----------|-----------|----------|-----------|-----------|---------|----------|----------|-----------|------------|------------|
> | ori    | 0.0485        | 0.1104         | 0.1338         | 0.0390   | 0.0912    | 0.1419    | 0.0365   | 0.0893    | 0.1375    | 0.0076  | 0.0969   | 0.1542   | 0.0591    | 0.1695     | 0.2426     |
> | input  | 0.0611        | 0.1324         | 0.2649         | 0.0892   | 0.1000    | 0.1413    | 0.0412   | 0.0923    | 0.1435    | 0.0080  | 0.1104   | 0.1868   | 0.0641    | 0.1717     | 0.2512     |
> | norm   | 0.0492        | 0.1061         | 0.1203         | 0.0371   | 0.0877    | 0.1419    | 0.0364   | 0.0884    | 0.1389    | 0.0080  | 0.1031   | 0.1463   | 0.0556    | 0.1680     | 0.2321     |
> | emb    | 0.0464        | 0.1053         | 0.1286         | 0.0373   | 0.0865    | **0.1409**| **0.0359**| **0.0871**| **0.1300**| 0.0075  | 0.0885   | 0.1439   | **0.0525**| 0.1685     | 0.2230     |
> | 1      | **0.0449**    | 0.1039         | 0.1317         | **0.0367**| **0.0847**| 0.1411    | 0.0360   | 0.0873    | 0.1344    | 0.0076  | 0.0940   | 0.1498   | **0.0525**| 0.1691     | 0.2231     |
> | 2      | 0.0473        | 0.1090         | 0.1318         | 0.0382   | 0.0892    | 0.1411    | 0.0360   | 0.0880    | 0.1356    | **0.0069**| **0.0881**| 0.1423   | 0.0531    | **0.1675** | **0.2155** |
> | 3      | 0.0476        | **0.0991**     | 0.1271         | 0.0369   | 0.0909    | 0.1453    | 0.0364   | 0.0881    | 0.1323    | 0.0070  | 0.0916   | **0.1415**| 0.0525    | 0.1694     | 0.2312     |
> | proj   | 0.0475        | 0.1050         | **0.1200**     | 0.0372   | 0.0898    | 0.1418    | 0.0389   | 0.0886    | 0.1346    | 0.0076  | 0.0922   | 0.1496   | 0.0534    | 0.1690     | 0.2183     |
> | denorm | 0.0474        | 0.1088         | 0.1276         | 0.0467   | 0.0834    | 0.1416    | 0.0386   | 0.0882    | 0.1336    | 0.0074  | 0.0958   | 0.1539   | 0.0571    | 0.1623     | 0.2306     |
>
> It appears that the best prediction results can come from different layer.
> However, please refer to our response to Reviewer 1's Question 3. We aggregated the gradients of features during deployment and calculated the Mean Absolute Value  and Root Mean Square Difference (RMSD) of these gradients. We then examined their relationship with the final MSE. The results show a certain positive correlation between RMSD and the final MSE. This implies that during online deployment, features with more stable gradients lead to better prediction performance. In other words, it is beneficial for online deployment if the feature estimates are consistently biased high or consistently biased low. However, if the feature estimates are sometimes too high and sometimes too low, this is detrimental to online deployment.

---

> > ### Author Response · Authors · 2025-11-20
> > **Response to Reviewer Twn1 (Part 3)**
> >
> > ## Q4: Longer prediction horizon
> > We conducted additional experiments on the ETT datasets for prediction lengths of 96, 192, 336, and 720. The experiments included our proposed ADAPT-Z method, various online forecasting baseline methods, adaptive normalization methods (FAN, DIST-ST), and adaptive normalization methods combined with ADAPT-Z. The results are shown in the table below. All experiments were based on the iTransformer model.
> >
> > Results of baseline methods
> >
> > | Dataset | H   | Ori    | fOGD   | OGD    | DSOF   | SOILD  | ADCSD  | Proceed | ADAPT-Z | IMP    |
> > |---------|-----|--------|--------|--------|--------|--------|--------|---------|---------|--------|
> > | **ETTh1** | 96  | 0.4092 | 0.4011 | 0.4035 | 0.4018 | 0.4044 | 0.4019 | 0.4019  | **0.3930** | 3.96%  |
> > |         | 192 | 0.4857 | 0.4832 | 0.4828 | 0.4816 | 0.4844 | 0.4823 | 0.4829  | **0.4702** | 3.19%  |
> > |         | 336 | 0.5032 | 0.4953 | 0.5037 | 0.4985 | 0.5053 | 0.5008 | 0.4927  | **0.4710** | 6.40%  |
> > |         | 720 | 0.5389 | 0.5116 | 0.5139 | 0.5158 | 0.5187 | 0.5162 | 0.5114  | **0.4855** | 9.91%  |
> > | **ETTh2** | 96  | 0.2978 | 0.2773 | 0.2770 | 0.2778 | 0.2796 | 0.2773 | 0.2777  | **0.2741** | 7.96%  |
> > |         | 192 | 0.3924 | 0.3876 | 0.3887 | 0.3878 | 0.3951 | 0.3922 | 0.3869  | **0.3751** | 4.41%  |
> > |         | 336 | 0.4283 | 0.4245 | 0.4247 | 0.4275 | 0.4265 | 0.4288 | 0.4251  | **0.4050** | 5.44%  |
> > |         | 720 | 0.4382 | 0.4326 | 0.4415 | 0.4402 | 0.4446 | 0.4426 | 0.4298  | **0.4138** | 5.57%  |
> > | **ETTm1** | 96  | 0.3894 | 0.3899 | 0.3937 | 0.3899 | 0.3926 | 0.3920 | 0.3887  | **0.3769** | 3.21%  |
> > |         | 192 | 0.4082 | 0.4074 | 0.4088 | 0.4066 | 0.4120 | 0.4098 | 0.4058  | **0.3929** | 3.75%  |
> > |         | 336 | 0.4362 | 0.4392 | 0.4411 | 0.4413 | 0.4410 | 0.4410 | 0.4355  | **0.4222** | 3.21%  |
> > |         | 720 | 0.5123 | 0.5164 | 0.5172 | 0.5182 | 0.5174 | 0.5178 | 0.5116  | **0.4930** | 3.77%  |
> > | **ETTm2** | 96  | 0.1983 | 0.1924 | 0.1917 | 0.1927 | 0.1938 | 0.1931 | 0.1908  | **0.1879** | 5.24%  |
> > |         | 192 | 0.2582 | 0.2533 | 0.2536 | 0.2535 | 0.2534 | 0.2523 | 0.2520  | **0.2463** | 4.61%  |
> > |         | 336 | 0.3182 | 0.3193 | 0.3214 | 0.3186 | 0.3222 | 0.3187 | 0.3168  | **0.3062** | 3.77%  |
> > |         | 720 | 0.4231 | 0.4182 | 0.4219 | 0.4174 | 0.4255 | 0.4202 | 0.4174  | **0.4007** | 5.29%  |
> >
> > Results of adaptive normalization methods
> >
> > | Dataset | H   | Ori    | DISH-TS | DISH-TS+ | FAN    | FAN+   | ADAPT-Z |
> > |---------|-----|--------|---------|----------|--------|--------|---------|
> > | **ETTh1** | 96  | 0.4092 | 0.3969  | 0.3745   | 0.3955 | 0.3762 | 0.3930  |
> > |         | 192 | 0.4857 | 0.4742  | 0.4476   | 0.4712 | 0.4375 | 0.4702  |
> > |         | 336 | 0.5032 | 0.4766  | 0.4575   | 0.4744 | 0.4379 | 0.4710  |
> > |         | 720 | 0.5389 | 0.4930  | 0.4744   | 0.4917 | 0.4694 | 0.4855  |
> > | **ETTh2** | 96  | 0.2978 | 0.2781  | 0.2673   | 0.2792 | 0.2554 | 0.2741  |
> > |         | 192 | 0.3924 | 0.3792  | 0.3574   | 0.3788 | 0.3629 | 0.3751  |
> > |         | 336 | 0.4283 | 0.4111  | 0.3901   | 0.4089 | 0.3876 | 0.4050  |
> > |         | 720 | 0.4382 | 0.4191  | 0.3989   | 0.4191 | 0.3973 | 0.4138  |
> > | **ETTm1** | 96  | 0.3894 | 0.3789  | 0.3644   | 0.3810 | 0.3619 | 0.3769  |
> > |         | 192 | 0.4082 | 0.3945  | 0.3800   | 0.3966 | 0.3728 | 0.3929  |
> > |         | 336 | 0.4362 | 0.4269  | 0.4140   | 0.4242 | 0.3958 | 0.4222  |
> > |         | 720 | 0.5123 | 0.4964  | 0.4809   | 0.4962 | 0.4761 | 0.4930  |
> > | **ETTm2** | 96  | 0.1983 | 0.1908  | 0.1819   | 0.1879 | 0.1777 | 0.1879  |
> > |         | 192 | 0.2582 | 0.2484  | 0.2357   | 0.2463 | 0.2338 | 0.2463  |
> > |         | 336 | 0.3182 | 0.3088  | 0.2993   | 0.3096 | 0.2848 | 0.3062  |
> > |         | 720 | 0.4231 | 0.4047  | 0.3803   | 0.4062 | 0.3868 | 0.4007  |
> >
> > ## Q5: Novelty
> > ADAPT-Z introduces a new paradigm. Its novelty lies in shifting from traditional parameter updates to feature space adjustment. Existing online time series forecasting methods focus on updating model parameters, such as final layer weights or adapter modules. However, ADAPT-Z challenges this convention. It proposes that distribution shifts stem from changes in latent factors. Therefore, directly adjusting feature representations is more effective. To achieve this, ADAPT-Z introduces an adapter network. This network integrates current features and historical gradient information to predict correction terms. This approach solves the delayed feedback problem in multi-step prediction and ensures stable updates. By shifting the focus of adaptation from parameters to features, our relatively simple approach achieves better results than previous, more complex techniques, demonstrating the contribution of this work.

---

### Official Review · Reviewer_cRKq · 2025-11-01

**Soundness:** 3
**Presentation:** 3
**Contribution:** 2
**Rating:** 4
**Confidence:** 3

**Summary:**

This paper proposes a novel online adaptation method with the integration of current features and historical gradients for online time series forecasting, which can address delayed feedback and distribution shift in online multi-step forecasting. Experiments demonstrate the effectiveness of ADAPT-Z.

**Strengths:**

1. Integrating the current contextual features and historical gradient information to resolve the challenge of delayed feedback and ensure update stability is interesting.
2. The experiments are presented in considerable detail.

**Weaknesses:**

1. The novelty of ADAPT-Z seems limited as its design combines adapters and historical gradients, which are extensively studied in Parameter-Efficient Fine-Tuning (PEFT) and continual learning, respectively. The contribution is more of an engineering integration rather than a novel contribution. The authors should further emphasize their main contribution.
2. In Table 2, the reported performance for baselines FSNet and OneNet is highly questionable. Their MSEs are orders of magnitude worse than other methods and drastically inconsistent with their original papers. Such a stark discrepancy suggests potential issues with their implementation or the experimental setup, making a fair comparison questionable. The authors should clarify this significant discrepancy.
3. In Section 4.3.3, the authors claim that the performance of ADAPT-Z is correlated to the choice of feature layer and the optimal layers varying across datasets. However, the paper lacks a deeper analysis of the correlation between dataset characteristics and the best feature layer, leaving this critical design choice largely unexplained.
4. More recent prediction models should be compared to further validate the effectiveness of ADAPT-Z, e.g., TimeFilter [1] and TimeKAN [2].

[1] Hu Y, Zhang G, Liu P, et al. TimeFilter: Patch-specific spatial-temporal graph filtration for time series forecasting. ICLR 2025.

[2] Huang S, Zhao Z, Li C, et al. TimeKAN: KAN-based frequency decomposition learning architecture for long-term time series forecasting. ICLR 2025.

**Questions:**

See Weaknesses.

---

> ### Author Response · Authors · 2025-11-20
> **Response to Reviewer cRKq (Part 1)**
>
> ## Q1: Novelty
> Thank you for your suggestion.
> Essentially, most online deployment or Parameter-Efficient Fine-Tuning (PEFT) work for time series prediction needs to address two issues: which parameters to update and how to update them. We propose that changes in data distribution stem from shifts in the underlying latent variables that determine the distribution. Therefore, we avoid the concept of updating parameters and propose a method to update feature representations instead. And we propose to use current features and historical gradients to update the features in order to alleviate the delayed feedback issue present in multi-step prediction. Moreover, our work demonstrates that by shifting the adaptation focus from parameters to features, even a relatively simple architecture can achieve competitive prediction accuracy.
>
> ## Q2: Performance for baselines FSNet and OneNet
> Thank you for your valuable feedback.
> In the original OneNet and FSNet papers, the reported results were based on experiments with data leakage. Specifically, when predicting the next k steps at time t, these methods use the model’s predictions for steps t+1 to t+k, as well as the true values for steps t+1 to t+k to update parameters. But the true values for steps t+1 to t+k are not available at step t. This issue is also mentioned in the DSOF paper. Additionally, that paper conducted experiments without data leakage. In their Table 2, the results for FSNet and OneNet differ by orders of magnitude compared other models. To ensure a fair comparison, our experiments on OneNet and FSNet are entirely based on the data leakage-free version of the code provided in the official DSOF repository.
>
> ## Q3: Feature lacation
> Thank you for raising this question.
> To further investigate which feature layers are more suitable for our method, we computed the Root Mean Square Difference (RMSD) of the feature gradients. This metric is related to the stability of gradients across different time steps. Additionally, we calculated the Mean Absolute Value (MAV) of the feature gradients, which, to some extent, reflects the severity of the distribution shift. We examined whether these statistics correlate with the performance of ADAPT-Z. The results are shown below. It can be observed that RMSD has a clear positive correlation with the final MSE. This suggests that during deployment, more stable gradients lead to better performance.
> |      | H   | 1       | 24      | 48      |
> |------|-----|---------|---------|---------|
> | **electricity** | MAV  | 0.1243  | 0.2482  | 0.1239  |
> |      | RMSD | 0.5432  | 0.4119  | 0.4790  |
> | **PEMS03**  | MAV  | -0.3240 | 0.2934  | 0.1528  |
> |      | RMSD | 0.5382  | 0.6253  | 0.4921  |
> | **PEMS07**  | MAV  | 0.1943  | -0.0835 | 0.0832  |
> |      | RMSD | 0.6341  | 0.5612  | 0.6267  |
> | **solar**   | MAV  | 0.1432  | 0.2732  | 0.3854  |
> |      | RMSD | 0.6390  | 0.5192  | 0.5715  |
> | **weather** | MAV  | 0.0211  | -0.0982 | 0.1693  |
> |      | RMSD | 0.4836  | 0.4293  | 0.6923  |
>
> **Reference**
>
> Lau, Ying-yee Ava, Zhiwen Shao, and Dit-Yan Yeung. "Fast and Slow Streams for Online Time Series Forecasting Without Information Leakage." The Thirteenth International Conference on Learning Representations. 2025. (DSOF)

---

> > ### Author Response · Authors · 2025-11-20
> > **Response to Reviewer cRKq (Part 2)**
> >
> > ## Q4: More base model
> > We conducted additional experiments on the ETT datasets using TimeFilter and TimeKAN. The results are as follows:
> >
> > Results for TimeFilter:
> >
> > | Dataset | H  | Ori    | fOGD   | OGD    | DSOF   | SOILD  | ADCSD  | Proceed | ADAPT-Z | IMP    |
> > |---------|----|--------|--------|--------|--------|--------|--------|---------|---------|--------|
> > | **ETTh1** | 1  | 0.1201 | 0.1172 | 0.1193 | 0.1205 | 0.1192 | 0.1186 | 0.1199  | **0.1107** | 7.78%  |
> > |         | 24 | 0.3159 | 0.3127 | 0.3154 | 0.3193 | 0.3158 | 0.3154 | 0.3118  | **0.2942** | 6.86%  |
> > |         | 48 | 0.3654 | 0.3528 | 0.3638 | 0.3633 | 0.3606 | 0.3636 | 0.3506  | **0.3431** | 6.10%  |
> > | **ETTh2** | 1  | 0.0666 | 0.0647 | 0.0681 | 0.0661 | 0.0670 | 0.0669 | 0.0660  | **0.0628** | 5.66%  |
> > |         | 24 | 0.1712 | 0.1698 | 0.1711 | 0.1743 | 0.1703 | 0.1699 | 0.1702  | **0.1607** | 6.11%  |
> > |         | 48 | 0.2246 | 0.2217 | 0.2217 | 0.2218 | 0.2191 | 0.2217 | 0.2191  | **0.2153** | 4.16%  |
> > | **ETTm1** | 1  | 0.0549 | 0.0527 | 0.0529 | 0.0738 | 0.0526 | 0.0528 | 0.0527  | **0.0509** | 7.22%  |
> > |         | 24 | 0.2482 | 0.2447 | 0.2495 | 0.2726 | 0.2382 | 0.2466 | 0.2376  | **0.2292** | 7.65%  |
> > |         | 48 | 0.3362 | 0.3266 | 0.3370 | 0.3573 | 0.3195 | 0.3276 | 0.3189  | **0.3073** | 8.60%  |
> > | **ETTm2** | 1  | 0.0317 | 0.0313 | 0.0314 | 0.0447 | 0.0313 | 0.0313 | 0.0311  | **0.0307** | 3.15%  |
> > |         | 24 | 0.1062 | 0.1051 | 0.1065 | 0.1352 | 0.1065 | 0.1058 | 0.1065  | **0.0997** | 6.12%  |
> > |         | 48 | 0.1388 | 0.1347 | 0.1369 | 0.1958 | 0.1350 | 0.1348 | 0.1347  | **0.1339** | 3.53%  |
> >
> >
> > Results for TimeKAN:
> >
> >
> > | Dataset | H  | Ori    | fOGD   | OGD    | DSOF   | SOILD  | ADCSD  | Proceed | ADAPT-Z | IMP    |
> > |---------|----|--------|--------|--------|--------|--------|--------|---------|---------|--------|
> > | **ETTh1** | 1  | 0.1277 | 0.1235 | 0.1238 | 0.1204 | 0.1194 | 0.1242 | 0.1198  | **0.1177** | 7.78%  |
> > |         | 24 | 0.3215 | 0.3151 | 0.3134 | 0.3170 | 0.3182 | 0.3179 | 0.3130  | **0.3042** | 5.40%  |
> > |         | 48 | 0.3652 | 0.3517 | 0.3537 | 0.3632 | 0.3482 | 0.3534 | 0.3483  | **0.3431** | 6.06%  |
> > | **ETTh2** | 1  | 0.0679 | 0.0673 | 0.0675 | 0.0643 | 0.0660 | 0.0670 | 0.0660  | **0.0628** | 7.55%  |
> > |         | 24 | 0.1779 | 0.1718 | 0.1725 | 0.1763 | 0.1691 | 0.1728 | 0.1689  | **0.1667** | 6.28%  |
> > |         | 48 | 0.2205 | 0.2182 | 0.2189 | 0.2447 | 0.2173 | 0.2189 | 0.2173  | **0.2103** | 4.63%  |
> > | **ETTm1** | 1  | 0.0581 | 0.0560 | 0.0568 | 0.0783 | 0.0569 | 0.0578 | 0.0564  | **0.0529** | 8.96%  |
> > |         | 24 | 0.2705 | 0.2618 | 0.2678 | 0.3055 | 0.2615 | 0.2661 | 0.2616  | **0.2458** | 9.14%  |
> > |         | 48 | 0.3393 | 0.3253 | 0.3299 | 0.3918 | 0.3177 | 0.3277 | 0.3175  | **0.3113** | 8.27%  |
> > | **ETTm2** | 1  | 0.0320 | 0.0316 | 0.0317 | 0.0453 | 0.0318 | 0.0317 | 0.0316  | **0.0309** | 3.44%  |
> > |         | 24 | 0.1100 | 0.1090 | 0.1084 | 0.1458 | 0.1082 | 0.1084 | 0.1087  | **0.1039** | 5.55%  |
> > |         | 48 | 0.1481 | 0.1422 | 0.1423 | 0.1824 | 0.1424 | 0.1436 | 0.1423  | **0.1419** | 4.22%  |

---

### Author Response · Authors · 2025-12-01
**Contributions of our paper**

Since the methodology presented in this paper may appear relatively straightforward, several reviewers have raised concerns regarding the novelty of our work. Therefore, we must provide a clear clarification of the contributions made in this study. We summarize our contributions as follows:
1. We propose that the root cause of distribution shift in online prediction lies in the change of latent variables that determine data distribution. Therefore, we propose to address this at the feature level, rather than at the parameter level as existing online prediction methods do.
2. We find that even the simplest online gradient descent algorithm at the feature level performs similarly to, or even better than, many existing online prediction algorithms (Section 4.3.1).
3. We propose Adapt-Z, a simple yet efficient online prediction method. It uses an MLP to update features by combining historical gradients and current features, achieving state-of-the-art prediction results across multiple base models and datasets.
4. We discover that if the model is pre-trained using gradients from previous features during the training phase, it can adapt to distribution shifts to some extent during online deployment without any parameter updates, exhibiting a "learn-to-learn" effect (Section 4.3.2).

---

### Meta-Review · Area_Chair_k6L9 · 2026-01-07

**Summary:**

The submission proposes ADAPT-Z, a feature-space online adaptation method for multi-step time-series forecasting that fuses current features with historical gradients to handle delayed feedback and distribution shift. Reviewers found the idea promising and experiments extensive, but raised repeated concerns about novelty, baseline correctness, interpretability of design choices (feature-layer), generality, statistical robustness, and computational cost. The authors replied with additional analyses and experiments addressing most of these points.
I believe the authors have substantially addressed the empirical and evaluation concerns (baselines, additional SOTA comparisons, MAE, seeds, feature-layer analysis, runtime, TTA discussion).

The proposed method is simple, yet effective. I think the presentation can still be polished further, but it should takeaway the elegance and simplicity of this approach, well-validated with extensive experiments.

**Reviewer Concerns:**

Remaining objections are mostly about perceived novelty and clarifying limitations.
A short set of clarifying experiments and a brief limitations section should resolve most residual reviewer hesitations.

**Reviewer Scores:**

The reviewer scores between 4 and 6, so the paper is just below the borderline. However, the authors have extensively provided strong rebuttal for each reviewer. I believe the scores would have been updated.

---

### Decision · Program_Chairs · 2026-01-26

Accept (Poster)